# DIRECTIONAL ANALYSIS OF STOCHASTIC GRADIENT DESCENT VIA VON MISES-FISHER DISTRIBUTIONS IN DEEP LEARNING

## ABSTRACT

Although stochastic gradient descent (SGD) is a driving force behind the recent success of deep learning, our understanding of its dynamics in a high-dimensional parameter space is limited. In recent years, some researchers have used the stochasticity of minibatch gradients, or the signal-to-noise ratio, to better characterize the learning dynamics of SGD. Inspired by these work, we here analyze SGD from a geometrical perspective by inspecting the stochasticity of the norms and directions of minibatch gradients. We propose a model of the directional concentration for minibatch gradients through von Mises-Fisher distribution and show that the directional uniformity of minibatch gradients increases over the course of SGD. We empirically verify our result using deep convolutional networks and observe a higher correlation between the gradient stochasticity and the proposed directional uniformity than that against the gradient norm stochasticity, suggesting that the directional statistics of minibatch gradients is a major factor behind SGD.

## 1 INTRODUCTION

Stochastic gradient descent (SGD) has been a driving force behind the recent success of deep learning. Despite a series of work on improving SGD by incorporating the second-order information of the objective function (Roux et al., 2008; Martens, 2010; Dauphin et al., 2014; Martens & Grosse, 2015; Desjardins et al., 2015), SGD is still the most widely used optimization algorithm for training a deep neural network. The learning dynamics of SGD, however, has not been well characterized beyond that it converges to an extremal point (Bottou, 1998) due to the non-convexity and high-dimensionality of a usual objective function used in deep learning.

Gradient stochasticity, or the signal-to-noise ratio (SNR) of the stochastic gradient, has been proposed as a tool for analyzing the learning dynamics of SGD. Shwartz-Ziv & Tishby (2017) identified two phases in SGD based on this. In the first phase, "drift phase", the gradient mean is much higher than its standard deviation, during which optimization progresses rapidly. This drift phase is followed by the "diffusion phase", where SGD behaves similarly to Gaussian noise with very small means. Similar observations were made by Li & Yuan (2017) and Chee & Toulis (2018) who have also divided the learning dynamics of SGD into two phases.

Shwartz-Ziv & Tishby (2017) have proposed that such phase transition is related to information compression. Unlike them, we notice that there are two aspects to the gradient stochasticity. One is the $L^2$ norm of the minibatch gradient (the norm stochasticity), and the other is the directional balance of minibatch gradients (the directional stochasticity). SGD converges or terminates when either the norm of the minibatch gradient vanishes to zeros, or when the angles of the minibatch gradients are uniformly distributed and their non-zero norms are close to each other. That is, the gradient stochasticity, or the SNR of the stochastic gradient, is driven by both of these aspects, and it is necessary for us to investigate not only the holistic SNR but also the SNR of the minibatch gradient norm and that of the minibatch gradient angles.

In this paper, we use a von Mises-Fisher (vMF hereafter) distribution, which is often used in directional statistics (Mardia & Jupp, 2009), and its concentration parameter $\kappa$ to characterize the directional balance of minibatch gradients and understand the learning dynamics of SGD from the perspective of directional statistics of minibatch gradients. We prove that SGD increases the direc-

tional balance of minibatch gradients. We empirically verify this with deep convolutional networks with various techniques, including batch normalization (Ioffe & Szegedy, 2015) and residual connections (He et al., 2015), on MNIST and CIFAR-10 (Krizhevsky & Hinton, 2009). Our empirical investigation further reveals that the proposed directional stochasticity is a major drive behind the gradient stochasticity compared to the norm stochasticity, suggesting the importance of understanding the directional statistics of the stochastic gradient.

**Contribution**   We analyze directional stochasticity of the minibatch gradients via angles as well as the concentration parameter of the vMF distribution. Especially, we theoretically show that the directional uniformity of the minibatch gradients modeled by the vMF distribution increases as training progresses, and verify this by experiments. In doing so, we introduce gradient norm stochasticity as the ratio of the standard deviation of the minibatch gradients to their expectation and theoretically and empirically show that this gradient norm stochasticity decreases as the batch size increases.

**Related work**   Most studies about SGD dynamics have been based on two-phase behavior (Shwartz-Ziv & Tishby, 2017; Li & Yuan, 2017; Chee & Toulis, 2018). Li & Yuan (2017) investigated this behavior by considering a shallow neural network with residual connections and assuming the standard normal input distribution. They showed that SGD-based learning under these setups has two phases; search and convergence phases. Shwartz-Ziv & Tishby (2017) on the other hand investigated a deep neural network with $\tanh$ activation functions, and showed that SGD-based learning has drift and diffusion phases. They have also proposed that such SNR transition (drift + diffusion) is related to the information transition divided into empirical error minimization and representation compression phases. However, Saxe et al. (2018) have reported that the information transition is not generally associated with the SNR transition with ReLU (Nair & Hinton, 2010) activation functions. Chee & Toulis (2018) instead looked at the inner product between successive minibatch gradients and presented transient and stationary phases.

Unlike our work here, the experimental verification of the previous work conducted under limited settings – the shallow network (Li & Yuan, 2017), the specific activation function (Shwartz-Ziv & Tishby, 2017), and only MNIST dataset (Shwartz-Ziv & Tishby, 2017; Chee & Toulis, 2018) – that conform well with their theoretical assumptions. Moreover, their work does not offer empirical result about the effect of the latest techniques including both batch normalization (Ioffe & Szegedy, 2015) layers and residual connections (He et al., 2015).

## 2   PRELIMINARIES

**Norms and Angles**   Unless explicitly stated, a norm refers to $L^2$ norm. $\| \cdot \|$ and $\langle \cdot, \cdot \rangle$ thus correspond to $L^2$ norm and the Euclidean inner product on $\mathbb{R}^d$, respectively. We use $\mathrm{x}_n \Rightarrow \mathrm{x}$ to indicate that "a random variable $\mathrm{x}_n$ converges to x in distribution." Similarly, $\mathrm{x}_n \xrightarrow{P} \mathrm{x}$ means convergence in probability. An angle $\theta$ between $d$-dimensional vectors $\boldsymbol{u}$ and $\boldsymbol{v}$ is defined by $\theta = \frac{180}{\pi} \cos^{-1} \left( \frac{\langle \boldsymbol{u}, \boldsymbol{v} \rangle}{\|\boldsymbol{u}\|\|\boldsymbol{v}\|} \right).$

**Loss functions**   A loss function of a neural network is written as $f(\boldsymbol{w}) = \frac{1}{n} \sum_{i=1}^{n} f_i(\boldsymbol{w})$, where $\boldsymbol{w} \in \mathbb{R}^d$ is a trainable parameter. $f_i$ is "a per-example loss function" computed on the $i$-th data point. We use $\mathbb{I}$ and $m$ to denote a minibatch index set and its batch size, respectively. Further, we call $f_{\mathbb{I}}(\boldsymbol{w}) = \frac{1}{m} \sum_{i \in \mathbb{I}} f_i(\boldsymbol{w})$ "a minibatch loss function given $\mathbb{I}$". In **Section 3.1**, we use $\boldsymbol{g}_i(\boldsymbol{w})$ and $\hat{\boldsymbol{g}}(\boldsymbol{w})$ to denote $-\nabla_{\boldsymbol{w}} f_i(\boldsymbol{w})$ and $-\nabla_{\boldsymbol{w}} f_{\mathbb{I}}(\boldsymbol{w})$, respectively. In **Section 3.3**, the index $i$ is used for the corresponding minibatch index set $\mathbb{I}_i$. For example, the negative gradient of $f_{\mathbb{I}_i}(\boldsymbol{w})$ is written as $\hat{\boldsymbol{g}}_i(\boldsymbol{w})$. During optimization, we denote a parameter $\boldsymbol{w}$ at the $i$-th iteration in the $t$-th epoch as $\boldsymbol{w}_t^i$, and $\boldsymbol{w}_0^0$ is an initial parameter. We use $n_b$ to refer to the number of minibatches in a single epoch.

**von Mises-Fisher Distribution**   We use the von Mises-Fisher (vMF) distribution to model the directions of vectors. The definition of the vMF distribution is as follows:

**Definition 1.** *(von Mises-Fisher Distribution, Banerjee et al. (2005)) The pdf of the* $\mathrm{vMF}(\boldsymbol{\mu}, \kappa)$ *is given by*

$$f_d(\boldsymbol{x}; \boldsymbol{\mu}, \kappa) = C_d(\kappa) \exp(\kappa \boldsymbol{\mu}^\top \boldsymbol{x})$$

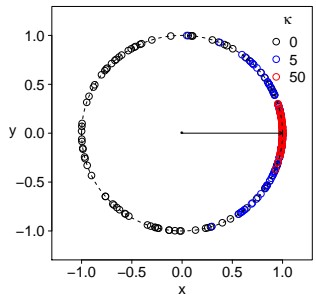

Figure 1: Characteristics of the vMF distribution in a 2-dimensional space. 100 random samples are drawn from $\text{vMF}(\boldsymbol{\mu}, \kappa)$ where $\boldsymbol{\mu} = (1, 0)^\top$ and $\kappa = \{0, 5, 50\}$.

*on the hypersphere $S^{d-1} \subset \mathbb{R}^d$. Here, the concentration parameter $\kappa$ determines how the samples from this distribution are concentrated on the mean direction $\boldsymbol{\mu}$ and $C_d(\kappa)$ is constant determined by $d$ and $\kappa$.*

If $\kappa$ is zero, then it is a uniform distribution on the unit hypersphere, and as $\kappa \to \infty$, it becomes a point mass on the unit hypersphere (Figure 1). The maximum likelihood estimates for $\boldsymbol{\mu}$ and $\kappa$ are $\hat{\boldsymbol{\mu}} = \frac{\sum_{i=1}^{n} \boldsymbol{x}_i}{\|\sum_{i=1}^{n} \boldsymbol{x}_i\|}$ and $\hat{\kappa} \approx \frac{\bar{r}(d - \bar{r}^2)}{1 - \bar{r}^2}$ where $\boldsymbol{x}_i$'s are random samples from the vMF distribution and $\bar{r} = \frac{\|\sum_{i=1}^{n} \boldsymbol{x}_i\|}{n}$. The formula for $\hat{\kappa}$ is approximate since the exact computation is intractable (Banerjee et al., 2005).

## 3 THEORETICAL MOTIVATION

### 3.1 ANALYSIS OF THE GRADIENT NORM STOCHASTICITY

It is a usual practice for SGD to use a minibatch gradient $\hat{\boldsymbol{g}}(\boldsymbol{w}) = -\nabla_{\boldsymbol{w}} f_{\mathbb{I}}(\boldsymbol{w})$ instead of a full batch gradient $\boldsymbol{g}(\boldsymbol{w}) = -\nabla_{\boldsymbol{w}} f(\boldsymbol{w})$. The minibatch index set $\mathbb{I}$ is drawn from $\{1, \ldots, n\}$ randomly. $\hat{\boldsymbol{g}}(\boldsymbol{w})$ satisfies $\mathbb{E}[\hat{\boldsymbol{g}}(\boldsymbol{w})] = \boldsymbol{g}(\boldsymbol{w})$ and $\text{Cov}(\hat{\boldsymbol{g}}(\boldsymbol{w}), \hat{\boldsymbol{g}}(\boldsymbol{w})) \approx \frac{1}{mn} \sum_{i=1}^{n} \boldsymbol{g}_i(\boldsymbol{w}) \boldsymbol{g}_i(\boldsymbol{w})^\top$ for $n \gg m$ where $n$ is the number of full data points and $\boldsymbol{g}_i(\boldsymbol{w}) = -\nabla_{\boldsymbol{w}} f_i(\boldsymbol{w})$ (Hoffer et al., 2017). As the batch size $m$ increases, the randomness of $\hat{\boldsymbol{g}}(\boldsymbol{w})$ decreases. Hence $\mathbb{E}\|\hat{\boldsymbol{g}}(\boldsymbol{w})\|$ tends to $\|\boldsymbol{g}(\boldsymbol{w})\|$, and $\text{Var}(\|\hat{\boldsymbol{g}}(\boldsymbol{w})\|)$, which is the variance of the norm of the minibatch gradient, vanishes. The convergence rate analysis is as the following:

**Theorem 1.** *Let $\hat{\boldsymbol{g}}(\boldsymbol{w})$ be a minibatch gradient induced from the minibatch index set $\mathbb{I}$ of batch size $m$ from $\{1, \ldots, n\}$ and suppose $\gamma = \max_{i,j \in \{1,\ldots,n\}} |\langle \boldsymbol{g}_i(\boldsymbol{w}), \boldsymbol{g}_j(\boldsymbol{w}) \rangle|$. Then*

$$0 \leq \mathbb{E}\|\hat{\boldsymbol{g}}(\boldsymbol{w})\| - \|\boldsymbol{g}(\boldsymbol{w})\| \leq \frac{2(n - m)}{m(n - 1)} \times \frac{\gamma}{\mathbb{E}\|\hat{\boldsymbol{g}}(\boldsymbol{w})\| + \|\boldsymbol{g}(\boldsymbol{w})\|} \leq \frac{(n - m)\gamma}{m(n - 1)\|\boldsymbol{g}(\boldsymbol{w})\|}$$

*and*

$$\text{Var}(\|\hat{\boldsymbol{g}}(\boldsymbol{w})\|) \leq \frac{2(n - m)}{m(n - 1)}\gamma \, .$$

*Hence,*

$$\frac{\sqrt{\text{Var}(\|\hat{\boldsymbol{g}}(\boldsymbol{w})\|)}}{\mathbb{E}\|\hat{\boldsymbol{g}}(\boldsymbol{w})\|} \leq \sqrt{\frac{2(n - m)}{m(n - 1)} \times \frac{\gamma}{\|\boldsymbol{g}(\boldsymbol{w})\|^2}} \, . \tag{1}$$

*Proof.* See **Supplemental A**. $\square$

According to **Theorem 1**, a large batch size $m$ reduces the variance of $\|\hat{\boldsymbol{g}}(\boldsymbol{w})\|$ centered at $\mathbb{E}\|\hat{\boldsymbol{g}}(\boldsymbol{w})\|$ with convergence rate $O(1/m)$. We empirically verify this by estimating the gradient norm stochasticity at random points while varying the minibatch size, using a fully-connected neural network (**FNN**) with MNIST, as shown in Figure 2(a) (see **Supplemental E** for more details.)

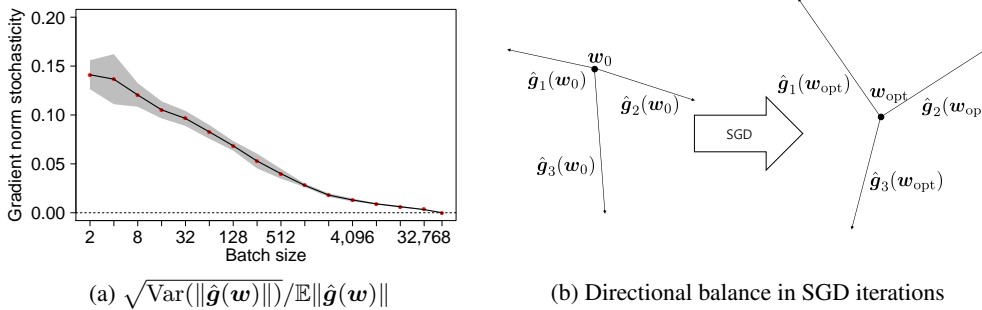

(a) $\sqrt{\mathrm{Var}(\|\hat{\boldsymbol{g}}(\boldsymbol{w})\|)}/\mathbb{E}\|\hat{\boldsymbol{g}}(\boldsymbol{w})\|$

(b) Directional balance in SGD iterations

Figure 2: The directions of minibatch gradients become more important than their lengths as the batch size increases. (a) The gradient norm stochasticity of $\hat{\boldsymbol{g}}(\boldsymbol{w})$ with respect to various batch sizes at 5 random points $\boldsymbol{w}$ with mean (black line) and mean±std.(shaded area) in a log-linear scale; (b) If the gradient norm stochasticity is sufficiently low, then the directions of $\hat{\boldsymbol{g}}_i(\boldsymbol{w})$'s need to be balanced to satisfy $\sum_{i=1}^{3} \hat{\boldsymbol{g}}_i(\boldsymbol{w}) \approx 0$.

This theorem however only demonstrate that the gradient norm stochasticity is (l.h.s. of (1)) is low at random initial points. It may blow up after SGD updates, since the upper bound (r.h.s. of (1)) is inversely proportional to $\|\boldsymbol{g}(\boldsymbol{w})\|$. This implies that the learning dynamics and convergence of SGD, measured in terms of the vanishing gradient, i.e., $\sum_{i=1}^{n_b} \hat{\boldsymbol{g}}_i(\boldsymbol{w}) \approx 0$, is not necessarily explained by the vanishing norms of minibatch gradients, but rather by the balance of the directions of $\hat{\boldsymbol{g}}_i(\boldsymbol{w})$'s, which motivates our investigation of the directional statistics of minibatch gradients. See Figure 2(b) as an illustration.

### 3.2 Uniformity Measurement via Analysis of Angles

In order to investigate the directions of minibatch gradients and how they balance, we start from an angle between two vectors. First, we analyze an asymptotic behavior of angles between uniformly random unit vectors in a high-dimensional space.

**Theorem 2.** *Suppose that* **u** *and* **v** *are mutually independent d-dimensional uniformly random unit vectors. Then,*

$$\sqrt{d} \times \left( \frac{180}{\pi} \cos^{-1} \langle \mathbf{u}, \mathbf{v} \rangle - 90 \right) \Rightarrow \mathcal{N}\left( 0, \left( \frac{180}{\pi} \right)^2 \right)$$

*as* $d \to \infty$.

*Proof.* See **Supplemental B**. $\square$

According to **Theorem 2**, the angle between two independent uniformly random unit vectors is normally distributed and becomes increasingly more concentrated as $d$ grows (Figure 3(a)). If SGD iterations indeed drive the directions of minibatch gradients to be uniform, then, at least, the distribution of angles between minibatch gradients and a given uniformly sampled unit vector follows asymptotically

$$\mathcal{N}\left( 90, \left( \frac{180}{\pi\sqrt{d}} \right)^2 \right). \tag{2}$$

Figures 3(b) and 3(c) show that the distribution of the angles between minibatch gradients and a given uniformly sampled unit vector converges to an asymptotic distribution (2) after SGD iterations. Although we could measure the uniformity of minibatch gradients how the angle distribution between minibatch gradients is close to (2), it is not as trivial to compare the distributions as to compare numerical values. This necessitates another way to measure the uniformity of minibatch gradients.

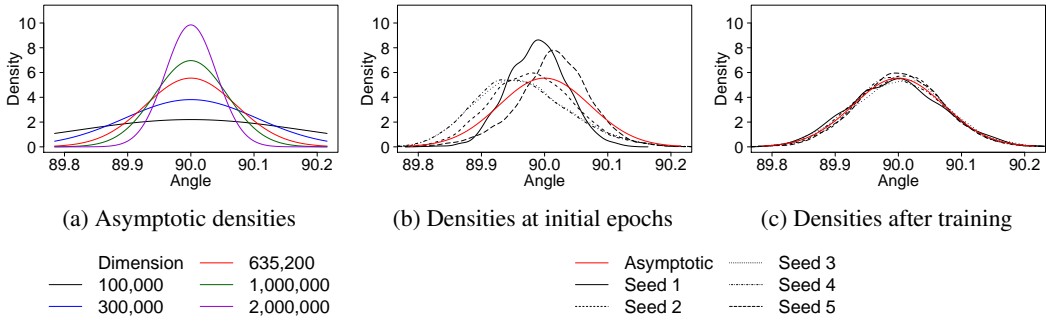

|  |  |  |
| --- | --- | --- |
| (a) Asymptotic densities | (b) Densities at initial epochs | (c) Densities after training |

Figure 3: (a) Asymptotic angle densities (2) of $\theta(\mathbf{u}, \mathbf{v}) = \frac{180}{\pi} \cos^{-1} \langle \mathbf{u}, \mathbf{v} \rangle$ where $\mathbf{u}$ and $\mathbf{v}$ are independent uniformly random unit vectors for each large dimension $d$. As $d \to \infty$, $\theta(\mathbf{u}, \mathbf{v})$ tends to less scattered from 90 (in degree). (b–c) We apply SGD on **FNN** for MNIST classification with the batch size 64 and the fixed learning rate 0.01 starting from five randomly initialized parameters. We draw a density plot $\theta(\boldsymbol{u}, \frac{\hat{\boldsymbol{g}}_j(\boldsymbol{w})}{\|\hat{\boldsymbol{g}}_j(\boldsymbol{w}))\|})$ for $3{,}000$ minibatch gradients (black) at $\boldsymbol{w} = \boldsymbol{w}_0^0$ (b) and $\boldsymbol{w} = \boldsymbol{w}_{\text{final}}^0$, with training accuracy of $> 99.9\%$, (c) when $\boldsymbol{u}$ is given. After SGD iterations, the density of $\theta(\boldsymbol{u}, \hat{\boldsymbol{g}}_j(\boldsymbol{w}))$ converges to an asymptotic density (red). The dimension of **FNN** is 635,200.

### 3.3 Uniformity measurement via vMF distribution

To model the uniformity of minibatch gradients, we propose to use the vMF distribution in **Definition 1**. The concentration parameter $\kappa$ measures how uniformly the directions of unit vectors are distributed. By **Theorem 1**, with a large batch size, the norm of minibatch gradient is nearly deterministic, and $\hat{\boldsymbol{\mu}}$ is almost parallel to the direction of full batch gradient. In other words, $\kappa$ measures the concentration of the minibatch gradients directions around the full batch gradient.

The following **Lemma 1** introduces the relationship between the norm of averaged unit vectors and $\hat{\kappa}$, the approximate estimator of $\kappa$.

**Lemma 1.** *The approximated estimator of $\kappa$ induced from the $d$-dimensional unit vectors* $\{\boldsymbol{x}_1, \boldsymbol{x}_2, \cdots, \boldsymbol{x}_{n_b}\}$,

$$\hat{\kappa} = \frac{\bar{r}(d - \bar{r}^2)}{1 - \bar{r}^2},$$

*is a strictly increasing function on $[0, 1]$, where $\bar{r} = \frac{\|\sum_{i=1}^{n_b} \boldsymbol{x}_i\|}{n_b}$. If we consider $\hat{\kappa} = h(u)$ as a function of $u = \|\sum_{i=1}^{n_b} \boldsymbol{x}_i\|$, then $h(\cdot)$ is Lipschitz continuous on $[0, n_b(1 - \epsilon)]$ for any $\epsilon > 0$. Moreover, $h(\cdot)$ and $h'(\cdot)$ are strictly increasing and increasing on $[0, n_b)$, respectively.*

*Proof.* See **Supplemental C.1**. ☐

Consider

$$\hat{\kappa}(\boldsymbol{w}) = h\left(\left\|\sum_{i=1}^{n_b} \frac{\boldsymbol{p}_i - \boldsymbol{w}}{\|\boldsymbol{p}_i - \boldsymbol{w}\|}\right\|\right),$$

which is measured from the directions from the current location $\boldsymbol{w}$ to the fixed points $\boldsymbol{p}_i$'s, where $h(\cdot)$ is a function defined in **Lemma 1**. Since $h(\cdot)$ is an increasing function, we may focus only on $\|\sum_{i=1}^{n_b} \frac{\boldsymbol{p}_i - \boldsymbol{w}}{\|\boldsymbol{p}_i - \boldsymbol{w}\|}\|$ to see how $\hat{\kappa}$ behaves with respect to its argument. **Lemma 2** implies that the estimated directional concentration $\hat{\kappa}$ decreases if we move away from $\boldsymbol{w}_0^0$ to $\boldsymbol{w}' = \boldsymbol{w}_0^0 + \epsilon \sum_i \frac{\boldsymbol{p}_i - \boldsymbol{w}_0^0}{\|\boldsymbol{p}_i - \boldsymbol{w}_0^0\|}$ with a small $\epsilon$ (Figure 4(a)). In other words, $\hat{\kappa}(\boldsymbol{w}') < \hat{\kappa}(\boldsymbol{w}_0^0)$.

**Lemma 2.** *Let $\boldsymbol{p}_1, \boldsymbol{p}_2, \cdots, \boldsymbol{p}_{n_b}$ be $d$-dimensional vectors. If all $\boldsymbol{p}_i$'s are not on a single ray from the current location $\boldsymbol{w}$, then there exists positive number $\eta$ such that*

$$\left\|\sum_{j=1}^{n_b} \frac{\boldsymbol{p}_j - \boldsymbol{w} - \epsilon \sum_{i=1}^{n_b} \frac{\boldsymbol{p}_i - \boldsymbol{w}}{\|\boldsymbol{p}_i - \boldsymbol{w}\|}}{\left\|\boldsymbol{p}_j - \boldsymbol{w} - \epsilon \sum_{i=1}^{n_b} \frac{\boldsymbol{p}_i - \boldsymbol{w}}{\|\boldsymbol{p}_i - \boldsymbol{w}\|}\right\|}\right\| < \left\|\sum_{i=1}^{n_b} \frac{\boldsymbol{p}_i - \boldsymbol{w}}{\|\boldsymbol{p}_i - \boldsymbol{w}\|}\right\|$$

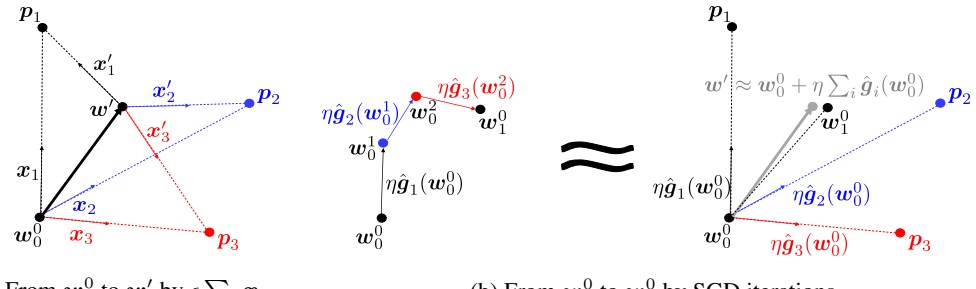

(a) From $\boldsymbol{w}_0^0$ to $\boldsymbol{w}'$ by $\epsilon \sum_i \boldsymbol{x}_i$  (b) From $\boldsymbol{w}_0^0$ to $\boldsymbol{w}_1^0$ by SGD iterations

Figure 4: (a) If the point is slightly moved from $\boldsymbol{w}_0^0$ to $\boldsymbol{w}'$ by $\epsilon \sum_i \boldsymbol{x}_i$ where $\boldsymbol{x}_i = (\boldsymbol{p}_i - \boldsymbol{w}_0^0)/\|\boldsymbol{p}_i - \boldsymbol{w}_0^0\|$ and $\boldsymbol{x}_i' = (\boldsymbol{p}_i - \boldsymbol{w}')/\|\boldsymbol{p}_i - \boldsymbol{w}'\|$, then $\|\sum_i \boldsymbol{x}_i'\| < \|\sum_i \boldsymbol{x}_i\|$ which is equivalent to $\hat{\kappa}(\boldsymbol{w}') < \hat{\kappa}(\boldsymbol{w}_0^0)$; (b) If $\hat{\boldsymbol{g}}_i(\boldsymbol{w}_0^0)$'s are sufficiently parallel to $(\boldsymbol{p}_i - \boldsymbol{w}_0^0)$'s for each $i$, then $\boldsymbol{w}' \approx \boldsymbol{w}_0^0 + \eta \sum_i \hat{\boldsymbol{g}}_i(\boldsymbol{w}_0^0)$. When $\boldsymbol{w}'$ and $\boldsymbol{w}_1^0$ are sufficiently close to each other, we also have $\hat{\kappa}(\boldsymbol{w}_1^0) < \hat{\kappa}(\boldsymbol{w}_0^0)$.

*for all $\epsilon \in (0, \eta]$.*

*Proof.* See **Supplemental C.2**. ☐

We make the connection between the observation above and SGD by first viewing $\boldsymbol{p}_i$'s as **local minibatch solutions**.

**Definition 2.** *For a minibatch index set $\mathbb{I}_i$, $\boldsymbol{p}_i(\boldsymbol{w}) = \arg\min_{\boldsymbol{w}' \in N(\boldsymbol{w}; r_i)} f_{\mathbb{I}_i}(\boldsymbol{w}')$ is a **local minibatch solution** of $\mathbb{I}_i$ at $\boldsymbol{w}$, where $N(\boldsymbol{w}; r_i)$ is a neighborhood of radius $r_i$ at $\boldsymbol{w}$. Here, $r_i$ is determined by $\boldsymbol{w}$ and $\mathbb{I}_i$ for $\boldsymbol{p}_i(\boldsymbol{w})$ to exist uniquely.*

Under this definition, $\boldsymbol{p}_i(\boldsymbol{w})$ is local minimum of a minibatch loss function $f_{\mathbb{I}_i}$ near $\boldsymbol{w}$. Then we reasonably expect that the direction of $\hat{\boldsymbol{g}}_i(\boldsymbol{w}) = -\nabla_{\boldsymbol{w}} f_{\mathbb{I}_i}(\boldsymbol{w})$ is similar to that of $\boldsymbol{p}_i(\boldsymbol{w}) - \boldsymbol{w}$.

Each epoch of SGD with a learning rate $\eta$ computes a series of $\boldsymbol{w}_t^j = \boldsymbol{w}_t^0 + \eta \sum_{i=1}^{j} \hat{\boldsymbol{g}}_i(\boldsymbol{w}_t^{i-1})$ for all $j \in \{1, \ldots, n_b\}$. If $\hat{\boldsymbol{g}}_i(\cdot)$'s are Lipschitz continuous for all $i \in \{1, \ldots, n_b\}$, then we have $\|\hat{\boldsymbol{g}}_i(\boldsymbol{w}_t^{i-1})\| \approx \|\hat{\boldsymbol{g}}_i(\boldsymbol{w}_t^0)\|$ for a small $\eta$. Moreover, **Theorem 1** implies $\|\hat{\boldsymbol{g}}_i(\boldsymbol{w}_t^0)\| \approx \tau$ for all $i \in \{1, \ldots, n_b\}$ with a large batch size or at the early stage of SGD iterations. Combining these approximations, $\|\hat{\boldsymbol{g}}_i(\boldsymbol{w}_t^{i-1})\| \approx \tau$ for all $i \in \{1, \ldots, n_b\}$.

For example, suppose that $t = 0$, $n_b = 3$ and $\tau = 1$, and assume that $\boldsymbol{p}_i(\boldsymbol{w}_0^0) = \boldsymbol{p}_i(\boldsymbol{w}_1^0) = \boldsymbol{p}_i$ for all $i = 1, 2, 3$. Then,

$$\hat{\kappa}(\boldsymbol{w}_0^0) = h\left(\left\|\sum_{i=1}^{3} \frac{\boldsymbol{p}_i - \boldsymbol{w}_0^0}{\|\boldsymbol{p}_i - \boldsymbol{w}_0^0\|}\right\|\right),$$

and

$$\hat{\kappa}(\boldsymbol{w}_1^0) \approx h\left(\left\|\sum_{j=1}^{3} \frac{\boldsymbol{p}_j - \boldsymbol{w}_0^0 - \eta \sum_{i=1}^{3} \frac{\hat{\boldsymbol{g}}_i(\boldsymbol{w}_0^{i-1})}{\|\hat{\boldsymbol{g}}_i(\boldsymbol{w}_0^{i-1})\|}}{\left\|\boldsymbol{p}_j - \boldsymbol{w}_0^0 - \eta \sum_{i=1}^{3} \frac{\hat{\boldsymbol{g}}_i(\boldsymbol{w}_0^{i-1})}{\|\hat{\boldsymbol{g}}_i(\boldsymbol{w}_0^{i-1})\|}\right\|}\right\|\right).$$

If $\eta \sum_i \frac{\boldsymbol{p}_i - \boldsymbol{w}_0^0}{\|\boldsymbol{p}_i - \boldsymbol{w}_0^0\|} \approx \eta \sum_i \frac{\hat{\boldsymbol{g}}_i(\boldsymbol{w}_0^{i-1})}{\|\hat{\boldsymbol{g}}_i(\boldsymbol{w}_0^{i-1})\|}$, then

$$\sum_{j=1}^{3} \frac{\boldsymbol{p}_j - \boldsymbol{w}_0^0 - \eta \sum_{i=1}^{3} \frac{\hat{\boldsymbol{g}}_i(\boldsymbol{w}_0^{i-1})}{\|\hat{\boldsymbol{g}}_i(\boldsymbol{w}_0^{i-1})\|}}{\left\|\boldsymbol{p}_j - \boldsymbol{w}_0^0 - \eta \sum_{i=1}^{3} \frac{\hat{\boldsymbol{g}}_i(\boldsymbol{w}_0^{i-1})}{\|\hat{\boldsymbol{g}}_i(\boldsymbol{w}_0^{i-1})\|}\right\|} \approx \sum_{j=1}^{3} \frac{\boldsymbol{p}_j - \boldsymbol{w}_0^0 - \eta \sum_{i=1}^{3} \frac{\boldsymbol{p}_i - \boldsymbol{w}_0^0}{\|\boldsymbol{p}_i - \boldsymbol{w}_0^0\|}}{\left\|\boldsymbol{p}_j - \boldsymbol{w}_0^0 - \eta \sum_{i=1}^{3} \frac{\boldsymbol{p}_i - \boldsymbol{w}_0^0}{\|\boldsymbol{p}_i - \boldsymbol{w}_0^0\|}\right\|}.$$

Hence, we have $\hat{\kappa}(\boldsymbol{w}_1^0) < \hat{\kappa}(\boldsymbol{w}_0^0)$ by **Lemma 2**. A trivial case satisfying this condition would be for each pair of $\boldsymbol{p}_i - \boldsymbol{w}_0^0$ and $\hat{\boldsymbol{g}}_i(\boldsymbol{w}_0^{i-1})$ to be approximately parallel, as illustrated in Figure 4(b).

**Theorem 3.** *Let $\boldsymbol{p}_1(\boldsymbol{w}_t^0), \boldsymbol{p}_2(\boldsymbol{w}_t^0), \cdots, \boldsymbol{p}_{n_b}(\boldsymbol{w}_t^0)$ be d-dimensional vectors, and all $\boldsymbol{p}_i(\boldsymbol{w}_t^0)$'s are not on a single ray from the current location $\boldsymbol{w}_t^0$. If*

$$\left\| \sum_{i=1}^{n_b} \frac{\boldsymbol{p}_i(\boldsymbol{w}_t^0) - \boldsymbol{w}_t^0}{\|\boldsymbol{p}_i(\boldsymbol{w}_t^0) - \boldsymbol{w}_t^0\|} - \sum_{i=1}^{n_b} \frac{\hat{\boldsymbol{g}}_i(\boldsymbol{w}_t^{i-1})}{\|\hat{\boldsymbol{g}}_i(\boldsymbol{w}_t^{i-1})\|} \right\| \leq \xi \tag{3}$$

*for a sufficiently small $\xi > 0$, then there exists positive number $\eta$ such that*

$$\left\| \sum_{j=1}^{n_b} \frac{\boldsymbol{p}_j(\boldsymbol{w}_t^0) - \boldsymbol{w}_t^0 - \epsilon \sum_{i=1}^{n_b} \frac{\hat{\boldsymbol{g}}_i(\boldsymbol{w}_t^{i-1})}{\|\hat{\boldsymbol{g}}_i(\boldsymbol{w}_t^{i-1})\|}}{\left\| \boldsymbol{p}_j(\boldsymbol{w}_t^0) - \boldsymbol{w}_t^0 - \epsilon \sum_{i=1}^{n_b} \frac{\hat{\boldsymbol{g}}_i(\boldsymbol{w}_t^{i-1})}{\|\hat{\boldsymbol{g}}_i(\boldsymbol{w}_t^{i-1})\|} \right\|} \right\| < \left\| \sum_{i=1}^{n_b} \frac{\boldsymbol{p}_i(\boldsymbol{w}_t^0) - \boldsymbol{w}_t^0}{\|\boldsymbol{p}_i(\boldsymbol{w}_t^0) - \boldsymbol{w}_t^0\|} \right\| \tag{4}$$

*for all $\epsilon \in (0, \eta]$.*

*Proof.* See **Supplemental C.3**. $\qquad\qquad\qquad\qquad\qquad\qquad\qquad\qquad\qquad\qquad\qquad\qquad\square$

This **Theorem 3** asserts that $\hat{\kappa}(\cdot)$ decreases even with some perturbation along the averaged direction $\sum_i \frac{\boldsymbol{p}_i(\boldsymbol{w}) - \boldsymbol{w}}{\|\boldsymbol{p}_i(\boldsymbol{w}) - \boldsymbol{w}\|}$. With additional assumptions on each minibatch loss functions, we have a sufficient condition for (3), summarized in **Corollary 3.1**.

**Corollary 3.1.** *Let $\boldsymbol{p}_i$ be the local minibatch solution of each $f_{\mathbb{I}_i}$. Suppose a region $\mathcal{R}$ satisfying:*

$$\text{For all } \boldsymbol{w}, \boldsymbol{w}' \in \mathcal{R}, \quad \boldsymbol{p}_i(\boldsymbol{w}) = \boldsymbol{p}_i(\boldsymbol{w}') = \boldsymbol{p}_i$$

*for all $i = 1, \cdots, n_b$. Further, assume that Hessian matrices of $f_{\mathbb{I}_i}$'s are positive definite, well-conditioned, and bounded in the sense of matrix $L^2$-norm on $\mathcal{R}$. If SGD moves from $\boldsymbol{w}_t^0$ to $\boldsymbol{w}_{t+1}^0$ on $\mathcal{R}$ with a large batch size and a small learning rate, then $\hat{\kappa}(\boldsymbol{w}_t^0) > \hat{\kappa}(\boldsymbol{w}_{t+1}^0)$. Moreover, we can estimate $\hat{\kappa}(\boldsymbol{w}_t^0)$ and $\hat{\kappa}(\boldsymbol{w}_{t+1}^0)$ by minibatch gradients at $\boldsymbol{w}_t^0$ and $\boldsymbol{w}_{t+1}^0$, respectively.*

*Proof.* See **Supplemental D**. $\qquad\qquad\qquad\qquad\qquad\qquad\qquad\qquad\qquad\qquad\qquad\qquad\qquad\square$

Without the corollary above, we need to solve $\boldsymbol{p}_i(\boldsymbol{w}_t^0) = \arg\min_{\boldsymbol{w} \in N(\boldsymbol{w}_t^0; r)} f_{\mathbb{I}_i}(\boldsymbol{w})$ for all $i \in \{1, \ldots, n_s\}$, where $n_s$ is the number of samples to estimate $\kappa$, in order to compute $\hat{\kappa}(\boldsymbol{w}_t^0)$. **Corollary 3.1** however implies that we can compute $\hat{\kappa}(\boldsymbol{w}_t^0)$ by using $\frac{\hat{\boldsymbol{g}}_i(\boldsymbol{w}_t^0)}{\|\hat{\boldsymbol{g}}_i(\boldsymbol{w}_t^0)\|}$ instead of $\frac{\boldsymbol{p}_i(\boldsymbol{w}_t^0) - \boldsymbol{w}_t^0}{\|\boldsymbol{p}_i(\boldsymbol{w}_t^0) - \boldsymbol{w}_t^0\|}$, significantly reducing computational overhead.

**In Practice** Although the number of all possible minibatches in each epoch is $n_b = \binom{n}{m}$, it is often the case to use $n_b' \approx n/m$ minibatches at each epoch in practice to go from $\boldsymbol{w}_t^0$ to $\boldsymbol{w}_{t+1}^0$. Assuming that these $n_b'$ minibatches were selected uniformly at random, the average of the $n_b'$ normalized minibatch gradients is the maximum likelihood estimate of $\boldsymbol{\mu}$, just like the average of all $n_b$ normalized minibatch gradients. Thus, we expect with a large $n_b'$,

$$\left\| \sum_{i=1}^{\binom{n}{m}} \frac{\boldsymbol{p}_i(\boldsymbol{w}_t^0) - \boldsymbol{w}_t^0}{\|\boldsymbol{p}_i(\boldsymbol{w}_t^0) - \boldsymbol{w}_t^0\|} - \sum_{i=1}^{n_b'} \frac{\hat{\boldsymbol{g}}_i(\boldsymbol{w}_t^{i-1})}{\|\hat{\boldsymbol{g}}_i(\boldsymbol{w}_t^{i-1})\|} \right\| \leq \xi,$$

and that SGD in practice also satisfies $\hat{\kappa}(\boldsymbol{w}_t^0) > \hat{\kappa}(\boldsymbol{w}_{t+1}^0)$.

## 4 EXPERIMENTS

### 4.1 SETUP

In order to empirically verify our theory on directional statistics of minibatch gradients, we train various types of deep neural networks using SGD and monitor the following metrics for analyzing the learning dynamics of SGD:

- Training loss

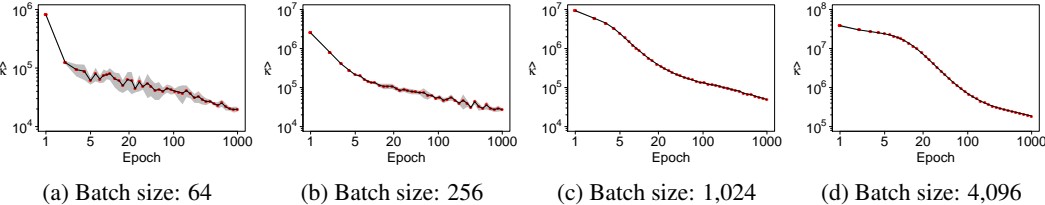

(a) Batch size: 64     (b) Batch size: 256     (c) Batch size: 1,024     (d) Batch size: 4,096

Figure 5: We show the average $\hat{\kappa}$ (black curve) $\pm$ std. (shaded area), as the function of the number of training epochs (in log-log scale) across various batch sizes in MNIST classifications using **FNN** with fixed learning rate $0.01$ and $5$ random initializations. Although $\hat{\kappa}$ with the large batch size decreases more smoothly rather than the small batch size, we observe that $\hat{\kappa}$ still decreases well with minibatches of size $64$. We did not match the ranges of the y-axes across the plots to emphasize the trend of monotonic decrease.

- Validation loss
- Gradient stochasticity (GS)$\uparrow$ $\|\mathbb{E}\nabla_{\boldsymbol{w}} f_{\mathbb{I}_i}(\boldsymbol{w})\|/\sqrt{\mathrm{tr}(\mathrm{Cov}(\nabla_{\boldsymbol{w}} f_{\mathbb{I}_i}(\boldsymbol{w}), \nabla_{\boldsymbol{w}} f_{\mathbb{I}_i}(\boldsymbol{w}))}$ $\downarrow$
- Gradient norm stochasticity (GNS) $\uparrow$ $\mathbb{E}\|\nabla_{\boldsymbol{w}} f_{\mathbb{I}_i}(\boldsymbol{w})\|/\sqrt{\mathrm{Var}(\|\nabla_{\boldsymbol{w}} f_{\mathbb{I}_i}(\boldsymbol{w})\|)}$ $\downarrow$
- Directional Uniformity$\uparrow$ $\kappa$ $\downarrow$

The latter three quantities are statistically estimated using $n_s = 3,000$ minibatches. We use $\hat{\kappa}$ to denote the $\kappa$ estimate.

We train the following types of deep neural networks (**Supplemental E**):

- **FNN**: a fully connected network with a single hidden layer
- **DFNN**: a fully connected network with three hidden layers
- **CNN**: a convolutional network with 14 layers (Krizhevsky et al., 2012)

In the case of the CNN, we also evaluate its variant with skip connections (**+Res**) (He et al., 2015). As it was shown recently by Santurkar et al. (2018) that batch normalization (Ioffe & Szegedy, 2015) improves the smoothness of a loss function in terms of its Hessian, we also test adding batch normalization to each layer right before the ReLU (Nair & Hinton, 2010) nonlinearity (**+BN**). We use MNIST for the FNN, DFNN and their variants, while CIFAR-10 (Krizhevsky & Hinton, 2009) for the CNN and its variants.

Our theory suggests a sufficiently large batch size for verification. We empirically analyze how large a batch size is needed in Figure 5. From these plots, $\hat{\kappa}$ decreases monotonically regardless of the minibatch size, but the variance over multiple training runs is much smaller with a larger minibatch size. We thus decide to use a practical size of $64$. With this fixed minibatch size, we use a fixed learning rate of $0.01$, which allows us to achieve the training accuracy of $> 99.9\%$ for every training run in our experiments. We repeat each setup five times starting from different random initial parameters and report both the mean and standard deviation.

## 4.2 DIRECTIONAL UNIFORMITY INCREASES

**FNN and DFNN** We first observe that $\hat{\kappa}$ decreases over training regardless of the network's depth in Figure 6 (a,b). We however also notice that $\hat{\kappa}$ decrease monotonically with the **FNN**, but less so with its deeper variant (**DFNN**). We conjecture this is due to the less-smooth loss landscape of a deep neural network. This difference between **FNN** and **DFNN** however almost entirely vanishes when batch normalization (**+BN**) is applied (Figure 6 (e,f)). This was expected as batch normalization is known to make the loss function behave better, and our theory assumes a smooth objective function.

**CNN** The **CNN** is substantially deeper than either **FNN** or **DFNN** and is trained on a substantially more difficult problem of CIFAR-10. In other words, the assumptions underlying our theory may not hold as well. Nevertheless, as shown in Figure 6 (c), $\hat{\kappa}$ eventually drops below its initial point, although this trend is not monotonic and $\hat{\kappa}$ fluctuates significantly over training. The addition of batch normalization (**+BN**) helps with the fluctuation but $\hat{\kappa}$ does not monotonically decrease

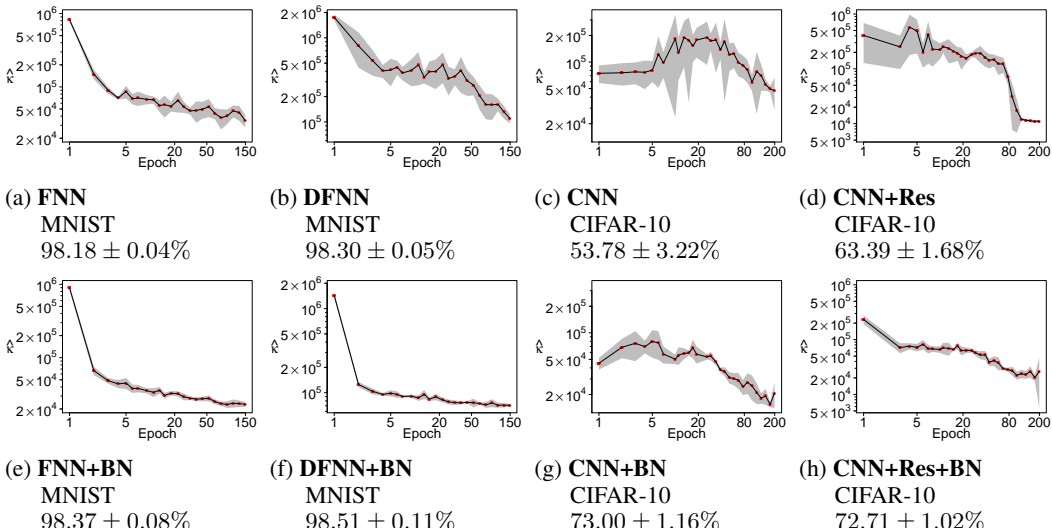

Figure 6: We show the average $\hat{\kappa}$ (black curve) $\pm$ std. (shaded area), as the function of the number of training epochs(in log-log scale) for all considered setups. We report the name of architecture, dataset and maximum valid accuracy(mean$\pm$std.) of all epochs. Although $\hat{\kappa}$ decreases eventually over time in all the cases, batch normalization (**+BN**) significantly reduces the variance of the directional stochasticity ((a–d) vs. (e–h)). We also observe that the skip connections make $\hat{\kappa}$ decrease monotonically ((c,g) vs. (d,h)). Note the differences in the y-scales.

(Figure 6 (g)). On the other hand, we observe the monotonic decrease of $\hat{\kappa}$ when skip connections (**+Res**) are introduced (Figure 6 (c) vs. Figure 6 (d)) albeit still with some level of fluctuation especially in the early stage of learning. When both batch normalization and skip connections are used (**+Res+BN**), the behaviour of $\hat{\kappa}$ matches with our prediction without much fluctuation.

**Effect of +BN and +Res** Based on our observations that the uniformity of minibatch gradients increases monotonically, when a deep neural network is equipped with residual connection (**+Res**) and trained with batch normalization (**+BN**), we conjecture that the loss function induced from these two techniques better satisfies the assumptions underlying our theoretical analysis, such as its well-behavedness. This conjecture is supported by for instance Santurkar et al. (2018), who demonstrated batch normalization guarantees the boundedness of Hessian, and Orhan & Pitkow (2017), who showed residual connections eliminate some singularities of Hessian.

$\hat{\kappa}$ **near the end of training** The minimum average $\hat{\kappa}$ of **DFNN+BN**, which has $1,920,000$ parameters, is $71,009.20$, that of **FNN+BN**, which has $636,800$ parameters, is $23,059.16$, and that of **CNN+BN+Res**, which has $207,152$ parameters, is $20,320.43$. These average $\hat{\kappa}$ are within a constant multiple of estimated $\kappa$ using 3,000 samples from the vMF distribution with true $\kappa = 0$ ($35,075.99$ with $1,920,000$ dimensions, $11,621.63$ with $636,800$ dimensions, and $3,781.04$ with $207,152$ dimensions.) This implies that we cannot say that the underlying directional distribution of minibatch gradients in all these cases at the end of training is not close to uniform (Cutting et al., 2017). For more detailed analysis, see **Supplementary F**.

### 4.3 DIRECTIONAL UNIFORMITY AND OTHER METRICS

The gradient stochasticity (GS) was used by Shwartz-Ziv & Tishby (2017) as a main metric for identifying two phases of SGD learning in deep neural networks. This quantity includes both the gradient norm stochasticity (GNS) and the directional uniformity $\kappa$, implying that either or both of GNS and $\kappa$ could drive the gradient stochasticity. We thus investigate the relationship among these three quantities as well training and validation losses. We focus on **CNN**, **CNN+BN** and **CNN+Res+BN** trained on CIFAR-10.

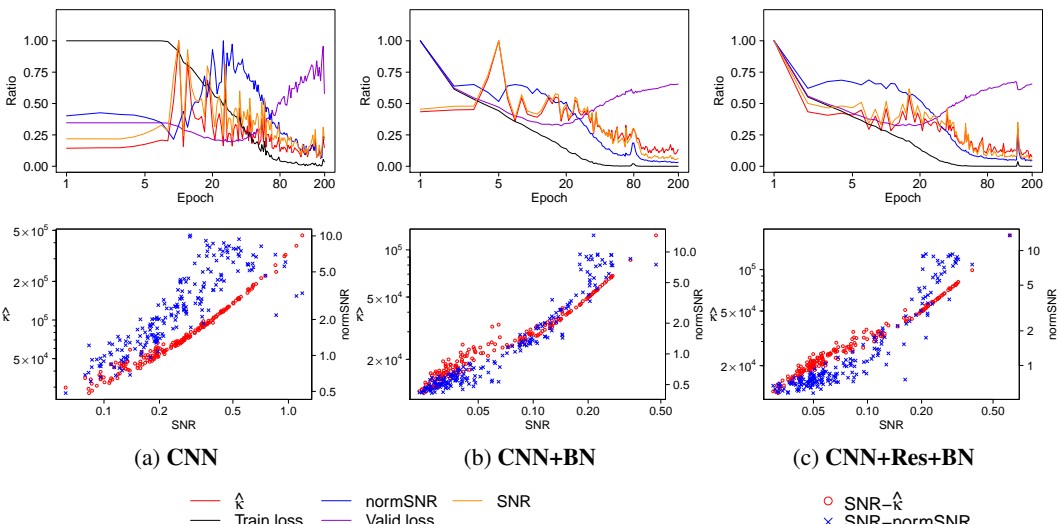

Figure 7: (First row) We plot the evolution of the training loss (Train loss), validation loss (Valid loss), inverse of gradient stochasticity (SNR), inverse of gradient norm stochasticity (normSNR) and directional uniformity $\hat{\kappa}$. We normalized each quantity by its maximum value over training for easier comparison on a single plot. In all the cases, SNR (orange) and $\hat{\kappa}$ (red) are almost entirely correlated with each other, while normSNR is less correlated. (Second row) We further verify this by illustrating SNR-$\hat{\kappa}$ scatter plots (red) and SNR-normSNR scatter plots (blue) in log-log scales. These plots suggest that the SNR is largely driven by the directional uniformity.

From Figure 7 (First row), it is clear that the proposed metric of directional uniformity $\hat{\kappa}$ correlates better with the gradient stochasticity than the gradient norm stochasticity does. This was especially prominent during the early stage of learning, suggesting that the directional statistics of minibatch gradients is a major explanatory factor behind the learning dynamics of SGD. This difference in correlations is much more apparent from the scatter plots in Figure 7 (Second row). We show these plots created from other four training runs per setup in **Supplemental G**.

## 5    CONCLUSION

Stochasticity of gradients is a key to understanding the learning dynamics of SGD (Shwartz-Ziv & Tishby, 2017) and has been pointed out as a factor behind the success of SGD (see, e.g., LeCun et al., 2012; Keskar et al., 2016). In this paper, we provide a theoretical framework using von Mises-Fisher distribution, under which the directional stochasticity of minibatch gradients can be estimated and analyzed, and show that the directional uniformity increases over the course of SGD. Through the extensive empirical evaluation, we have observed that the directional uniformity indeed improves over the course of training a deep neural network, and that its trend is monotonic when batch normalization and skip connections were used. Furthermore, we demonstrated that the stochasticity of minibatch gradients is largely determined by the directional stochasticity rather than the gradient norm stochasticity.

Our work in this paper suggests two major research directions for the future. First, our analysis has focused on the aspect of optimization, and it is an open question how the directional uniformity relates to the generalization error although handling the stochasticity of gradients has improved SGD (Neelakantan et al., 2015; Hoffer et al., 2017; Smith et al., 2017; Jin et al., 2017). Second, we have focused on passive analysis of SGD using the directional statistics of minibatch gradients, but it is not unreasonable to suspect that SGD could be improved by explicitly taking into account the directional statistics of minibatch gradients during optimization.

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

## Supplementary Material

## A  Proofs for Theorem 1

In proving **Theorem 1**, we use **Lemma A.1**. Define selector random variables(Hoffer et al., 2017) as below:

$$
\mathrm{s}_i = \left\{ \begin{array}{ll} 1, & \text{if } i \in \mathbb{I} \\ 0, & \text{if } i \notin \mathbb{I} \end{array} \right. .
$$

Then we have

$$
\hat{\boldsymbol{g}}(\boldsymbol{w}) = \frac{1}{m} \sum_{i=1}^{n} \boldsymbol{g}_i(\boldsymbol{w}) \mathrm{s}_i.
$$

**Lemma A.1.** *Let $\hat{\boldsymbol{g}}(\boldsymbol{w})$ be a minibatch gradient induced from the minibatch index set $\mathbb{I}$ with batch size $m$ from $\{1, \ldots, n\}$. Then*

$$
0 \le \mathbb{E}\|\hat{\boldsymbol{g}}(\boldsymbol{w})\|^2 - \|\boldsymbol{g}(\boldsymbol{w})\|^2 \le \frac{2(n-m)}{m(n-1)} \gamma. \tag{5}
$$

*where $\gamma = \max_{i,j\in\{1,\ldots,n\}} |\langle \boldsymbol{g}_i(\boldsymbol{w}), \boldsymbol{g}_j(\boldsymbol{w}) \rangle|$.*

*Proof.* By Jensen's inequality, $0 \le \mathbb{E}\|\hat{\boldsymbol{g}}(\boldsymbol{w})\|^2 - \|\boldsymbol{g}(\boldsymbol{w})\|^2$. Note that

$$
\mathbb{E}\|\hat{\boldsymbol{g}}(\boldsymbol{w})\|^2 = \sum_{i=1}^{n} \sum_{j=1}^{n} \frac{1}{m^2} \langle \boldsymbol{g}_i(\boldsymbol{w}), \boldsymbol{g}_j(\boldsymbol{w}) \rangle \mathbb{E}[\mathrm{s}_i \mathrm{s}_j].
$$

Since $\mathbb{E}[\mathrm{s}_i \mathrm{s}_j] = \frac{m}{n}\delta_{ij} + \frac{m(m-1)}{n(n-1)}(1 - \delta_{ij})$,

$$
\begin{aligned}
\mathbb{E}\|\hat{\boldsymbol{g}}(\boldsymbol{w})\|^2 - \|\boldsymbol{g}(\boldsymbol{w})\|^2 &= \Big(\frac{1}{mn} - \frac{m-1}{mn(n-1)}\Big) \sum_{i=1}^{n} \langle \boldsymbol{g}_i(\boldsymbol{w}), \boldsymbol{g}_i(\boldsymbol{w}) \rangle \\
&\quad + \Big(\frac{m-1}{mn(n-1)} - \frac{1}{n^2}\Big) \sum_{i=1}^{n} \sum_{j=1}^{n} \langle \boldsymbol{g}_i(\boldsymbol{w}), \boldsymbol{g}_j(\boldsymbol{w}) \rangle \\
&= \Big(\frac{1}{mn} - \frac{m-1}{mn(n-1)}\Big) \sum_{i=1}^{n} \langle \boldsymbol{g}_i(\boldsymbol{w}), \boldsymbol{g}_i(\boldsymbol{w}) \rangle \\
&\quad + \frac{m-n}{mn^2(n-1)} \sum_{i=1}^{n} \sum_{j=1}^{n} \langle \boldsymbol{g}_i(\boldsymbol{w}), \boldsymbol{g}_j(\boldsymbol{w}) \rangle \\
&\le \Big(\frac{1}{mn} - \frac{m-1}{mn(n-1)}\Big) n\gamma + \frac{n-m}{mn^2(n-1)} n^2 \gamma \\
&= \frac{2(n-m)}{m(n-1)} \gamma
\end{aligned}
$$

where $\gamma = \max_{i,j\in\{1,\ldots,n\}} |\langle \boldsymbol{g}_i(\boldsymbol{w}), \boldsymbol{g}_j(\boldsymbol{w}) \rangle|$. $\qquad\square$

**Theorem 1.** *Let $\hat{\boldsymbol{g}}(\boldsymbol{w})$ be a minibatch gradient induced from the minibatch index set $\mathbb{I}$ of batch size $m$ from $\{1, \ldots, n\}$ and suppose $\gamma = \max_{i,j\in\{1,\ldots,n\}} |\langle \boldsymbol{g}_i(\boldsymbol{w}), \boldsymbol{g}_j(\boldsymbol{w}) \rangle|$. Then*

$$
0 \le \mathbb{E}\|\hat{\boldsymbol{g}}(\boldsymbol{w})\| - \|\boldsymbol{g}(\boldsymbol{w})\| \le \frac{2(n-m)}{m(n-1)} \times \frac{\gamma}{\mathbb{E}\|\hat{\boldsymbol{g}}(\boldsymbol{w})\| + \|\boldsymbol{g}(\boldsymbol{w})\|} \le \frac{(n-m)\gamma}{m(n-1)\|\boldsymbol{g}(\boldsymbol{w})\|}
$$

*and*

$$
\mathrm{Var}(\|\hat{\boldsymbol{g}}(\boldsymbol{w})\|) \le \frac{2(n-m)}{m(n-1)} \gamma .
$$

*Hence,*

$$
\frac{\sqrt{\mathrm{Var}(\|\hat{\boldsymbol{g}}(\boldsymbol{w})\|)}}{\mathbb{E}\|\hat{\boldsymbol{g}}(\boldsymbol{w})\|} \le \sqrt{\frac{2(n-m)}{m(n-1)} \times \frac{\gamma}{\|\boldsymbol{g}(\boldsymbol{w})\|^2}} .
$$

*Proof.* By Jensen's inequality, we have $\|g(w)\| = \|\mathbb{E}[\hat{g}(w)]\| \le \mathbb{E}\|\hat{g}(w)\|$ and $(\mathbb{E}\|\hat{g}(w)\|)^2 \le \mathbb{E}\|\hat{g}(w)\|^2$. From the second inequality and **Lemma A.1**,

$$(\mathbb{E}\|\hat{g}(w)\|)^2 \le \mathbb{E}\|\hat{g}(w)\|^2 \le \|g(w)\|^2 + \frac{2(n-m)}{m(n-1)}\gamma$$

or

$$\big(\mathbb{E}\|\hat{g}(w)\| - \|g(w)\|\big)\big(\mathbb{E}\|\hat{g}(w)\| + \|g(w)\|\big) \le \frac{2(n-m)}{m(n-1)}\gamma \ .$$

Hence

$$\mathbb{E}\|\hat{g}(w)\| \le \|g(w)\| + \frac{2(n-m)}{m(n-1)} \times \frac{\gamma}{\mathbb{E}\|\hat{g}(w)\| + \|g(w)\|} \le \frac{(n-m)\gamma}{m(n-1)\|g(w)\|}.$$

Further,

$$\begin{aligned}
\mathrm{Var}(\|\hat{g}(w)\|) &= \mathbb{E}\|\hat{g}(w)\|^2 - (\mathbb{E}\|\hat{g}(w)\|)^2 \\
&\le \mathbb{E}\|\hat{g}(w)\|^2 - \|\mathbb{E}\hat{g}(w)\|^2 \\
&\le \|g(w)\|^2 + \frac{2(n-m)}{m(n-1)}\gamma - \|g(w)\|^2 = \frac{2(n-m)}{m(n-1)}\gamma.
\end{aligned}$$

$\square$

# B   PROOFS FOR THEOREM 2

For proofs, Slutsky's theorem and delta method are key results to describe limiting behaviors of random variables in distributional sense.

**Theorem B.1.** *(Slutsky's theorem, Casella & Berger (2002)) Let $\{x_n\}$, $\{y_n\}$ be a sequence of random variables that satisfies $x_n \Rightarrow x$ and $y_n \xrightarrow{P} \rho$ when $n$ goes to infinity and $\rho$ is constant. Then*

$$x_n y_n \Rightarrow cx$$

**Theorem B.2.** *(Delta method, Casella & Berger (2002)) Let $y_n$ be a sequence of random variables that satisfies $\sqrt{n}(y_n - \mu) \Rightarrow \mathcal{N}(0, \sigma^2)$. For a given smooth function $f : \mathbb{R} \to \mathbb{R}$, suppose that $f'(\mu)$ exists and is not 0 where $f'$ is a derivative. Then*

$$\sqrt{n} \times (f(y_n) - f(\mu)) \Rightarrow \mathcal{N}(0, \sigma^2(f'(\mu))^2).$$

**Lemma B.1.** *Suppose that $u$ and $v$ are mutually independent $d$-dimensional uniformly random unit vectors. Then, $\sqrt{d}\langle u, v \rangle \Rightarrow \mathcal{N}(0, 1)$ as $d \to \infty$.*

*Proof.* Note that $d$-dimensional uniformly random unit vectors $u$ can be generated by normalization of $d$-dimensional multivariate standard normal random vectors $x \sim N(0, I_d)$. That is,

$$u \sim \frac{x}{\|x\|}.$$

Suppose that two independent uniformly random unit vector $u$ and $v$ are generated by two independent $d$-dimensional standard normal vector $x = (x_1, x_2, \cdots, x_d)$ and $y = (y_1, y_2, \cdots, y_d)$. Denote them

$$u = \frac{x}{\|x\|} \quad \text{and} \quad v = \frac{y}{\|y\|}.$$

By SLLN, we have

$$\frac{\|x\|}{\sqrt{d}} \to 1 \quad a.s.$$

(Use $\frac{1}{d}\sum_{i=1}^{d} x_i^2 \to \mathbb{E}x_1^2 = 1$). Since almost sure convergence implies convergence in probability, $\|x\|/\sqrt{d} \xrightarrow{P} 1$. Similarly, $\|y\|/\sqrt{d} \xrightarrow{P} 1$. Moreover, by CLT,

$$\frac{\langle x, y \rangle}{\sqrt{d}} = \sqrt{d}\Big(\frac{1}{d}\sum_{i=1}^{d} x_i y_i\Big) \Rightarrow \mathcal{N}(0, 1).$$

Therefore, by **Theorem B.1** (Slutsky's theorem),

$$\sqrt{d}\langle u, v \rangle \Rightarrow \mathcal{N}(0, 1).$$

$\square$

**Theorem 2.** *Suppose that* **u** *and* **v** *are mutually independent d-dimensional uniformly random unit vectors. Then,*

$$\sqrt{d} \times \left( \frac{180}{\pi} \cos^{-1} \langle \mathbf{u}, \mathbf{v} \rangle - 90 \right) \Rightarrow \mathcal{N}\left( 0, \left( \frac{180}{\pi} \right)^2 \right)$$

*as* $d \to \infty$.

*Proof.* Suppose that $\mu = 0$, $\sigma = 1$, and $f(\cdot) = \frac{180}{\pi} \cos^{-1}(\cdot)$. Since $\frac{180}{\pi} \frac{d}{dx} \cos^{-1}(x) = -\frac{180}{\pi\sqrt{1-x^2}}$, we have $f'(\mu) = -\frac{180}{\pi}$. Hence, by **Lemma B.1** and **Theorem B.2** (Delta method), the desired convergence in distribution holds. $\square$

## C   PROOFS FOR THEOREM 3

### C.1   PROOF OF LEMMA 1

**Lemma 1.** *The approximated estimator of* $\kappa$ *induced from the d-dimensional unit vectors* $\{\boldsymbol{x}_1, \boldsymbol{x}_2, \cdots, \boldsymbol{x}_{n_b}\}$,

$$\hat{\kappa} = \frac{\bar{r}(d - \bar{r}^2)}{1 - \bar{r}^2},$$

*where* $\bar{r} = \frac{\|\sum_{i=1}^{n_b} \boldsymbol{x}_i\|}{n_b}$ *is a strict increasing function on* $[0, 1]$. *If we consider* $\hat{\kappa} = h(u)$ *as function of* $u = \|\sum_{i=1}^{n_b} \boldsymbol{x}_i\|$. *Then* $h(\cdot)$ *is Lipschitz continuous on* $[0, n_b(1 - \epsilon)]$ *for any* $\epsilon > 0$. *Moreover,* $h(\cdot)$ *and* $h'(\cdot)$ *are strict increasing and increasing on* $[0, n_b)$, *respectively.*

*Proof.* Note that $\|\sum_{i=1}^{n_b} \boldsymbol{x}_i\| \leq \sum_{i=1}^{n_b} \|\boldsymbol{x}_i\| = n_b$. Therefore, we have $\bar{r} \in [0, 1]$. If $d = 1$, then $\hat{\kappa} = \bar{r}$ and this increases on $[0, 1]$. For $d > 1$,

$$\frac{d\hat{\kappa}}{d\bar{r}} = \frac{d + \bar{r}^4 + (d - 3\bar{r}^2)}{(1 - \bar{r}^2)^2}$$

and its numerator is always positive for $d > 2$. When $d = 2$,

$$\frac{d\hat{\kappa}}{d\bar{r}} = \frac{\bar{r}^4 - 3\bar{r}^2 + 4}{(1 - \bar{r}^2)^2} = \frac{(\bar{r}^2 - \frac{3}{2})^2 + \frac{7}{4}}{(1 - \bar{r}^2)^2} > 0.$$

So $\hat{\kappa}$ increases as $\bar{r}$ increases.

The Lipschitz continuity of $h(\cdot)$ directly comes from the continuity of $\frac{d\hat{\kappa}}{d\bar{r}}$ since

$$\frac{d\hat{\kappa}}{du} = \frac{1}{n_b} \frac{d\hat{\kappa}}{d\bar{r}}.$$

Recall that any continuous function on the compact interval $[0, n_b(1 - \epsilon)]$ is bounded. Hence the derivative of $\hat{\kappa}$ with respect to $u$ is bounded. This implies the Lipschitz continuity of $h(\cdot)$.

$h(\cdot)$ is strictly increasing since $\bar{r} = \frac{u}{n_b}$. Further,

$$h''(u) = \frac{1}{n_b^2} \frac{d^2\hat{\kappa}}{d\bar{r}^2}$$

$$= \frac{2\bar{r}^5 + (4 - 8d)\bar{r}^3 + (8d - 6)\bar{r}}{n_b^2(1 - \bar{r}^2)^4} > 0$$

due to $\bar{r} \in [0, 1]$. Therefore $h'(\cdot)$ is also increasing on $[0, n_b)$. $\square$

### C.2   PROOF OF LEMMA 2

**Lemma 2.** *Let* $\boldsymbol{p}_1, \boldsymbol{p}_2, \cdots, \boldsymbol{p}_{n_b}$ *be d-dimensional vectors. If all* $\boldsymbol{p}_i$*'s are not on a single ray from the current location* $\boldsymbol{w}$, *then there exists positive number* $\eta$ *such that*

$$\left\| \sum_{j=1}^{n_b} \frac{\boldsymbol{p}_j - \boldsymbol{w} - \epsilon \sum_{i=1}^{n_b} \frac{\boldsymbol{p}_i - \boldsymbol{w}}{\|\boldsymbol{p}_i - \boldsymbol{w}\|}}{\|\boldsymbol{p}_j - \boldsymbol{w} - \epsilon \sum_{i=1}^{n_b} \frac{\boldsymbol{p}_i - \boldsymbol{w}}{\|\boldsymbol{p}_i - \boldsymbol{w}\|}\|} \right\| < \left\| \sum_{i=1}^{n_b} \frac{\boldsymbol{p}_i - \boldsymbol{w}}{\|\boldsymbol{p}_i - \boldsymbol{w}\|} \right\|$$

*for all* $\epsilon \in (0, \eta]$.

*Proof.* Without loss of generality, we regard $\boldsymbol{w}$ as the origin. Let $f(\epsilon) = \left\| \sum_{j=1}^{n_b} \frac{\boldsymbol{p}_j - \epsilon \sum_{i=1}^{n_b} \frac{\boldsymbol{p}_i}{\|\boldsymbol{p}_i\|}}{\|\boldsymbol{p}_j - \epsilon \sum_{i=1}^{n_b} \frac{\boldsymbol{p}_i}{\|\boldsymbol{p}_i\|}\|} \right\|^2$, then $f(0) = \left\| \sum_{i=1}^{n_b} \frac{\boldsymbol{p}_i}{\|\boldsymbol{p}_i\|} \right\|^2$. Therefore, we only need to show $f'(0) < 0$. Now denote $\boldsymbol{x}_j = \frac{\boldsymbol{p}_j}{\|\boldsymbol{p}_j\|}$, $\boldsymbol{p}_j(\epsilon) = \boldsymbol{p}_j - \epsilon \sum_{i=1}^{n_b} \boldsymbol{x}_i$ and $\boldsymbol{u} = -\sum_{i=1}^{n_b} \boldsymbol{x}_i$. That is, $\boldsymbol{p}_j(\epsilon) = \boldsymbol{p}_j + \epsilon \boldsymbol{u}$. Since

$$f(\epsilon) = \Big\langle \sum_{j=1}^{n_b} \frac{\boldsymbol{p}_j(\epsilon)}{\|\boldsymbol{p}_j(\epsilon)\|}, \sum_{j=1}^{n_b} \frac{\boldsymbol{p}_j(\epsilon)}{\|\boldsymbol{p}_j(\epsilon)\|} \Big\rangle,$$

we have

$$f'(\epsilon) = 2 \Big\langle \sum_{j=1}^{n_b} \frac{\boldsymbol{p}_j(\epsilon)}{\|\boldsymbol{p}_j(\epsilon)\|}, \frac{d}{d\epsilon}\Big( \sum_{j=1}^{n_b} \frac{\boldsymbol{p}_j(\epsilon)}{\|\boldsymbol{p}_j(\epsilon)\|} \Big) \Big\rangle$$

and

$$\frac{d}{d\epsilon}\Big( \sum_{j=1}^{n_b} \frac{\boldsymbol{p}_j(\epsilon)}{\|\boldsymbol{p}_j(\epsilon)\|} \Big) = \sum_{j=1}^{n_b} \frac{\|\boldsymbol{p}_j(\epsilon)\|\boldsymbol{u} - \frac{\langle \boldsymbol{u}, \boldsymbol{p}_j(\epsilon)\rangle}{\|\boldsymbol{p}_j(\epsilon)\|}\boldsymbol{p}_j(\epsilon)}{\|\boldsymbol{p}_j(\epsilon)\|^2}.$$

Hence

$$f'(\epsilon) = 2 \Big\langle \sum_{j=1}^{n_b} \frac{\boldsymbol{p}_j(\epsilon)}{\|\boldsymbol{p}_j(\epsilon)\|}, \sum_{j=1}^{n_b} \frac{\|\boldsymbol{p}_j(\epsilon)\|\boldsymbol{u} - \frac{\langle \boldsymbol{u}, \boldsymbol{p}_j(\epsilon)\rangle}{\|\boldsymbol{p}_j(\epsilon)\|}\boldsymbol{p}_j(\epsilon)}{\|\boldsymbol{p}_j(\epsilon)\|^2} \Big\rangle.$$

Note that $\boldsymbol{p}_j(0) = \boldsymbol{p}_j$ and $\|\boldsymbol{x}_j\| = 1$. We have

$$f'(0) = 2 \Big\langle \sum_{j=1}^{n_b} \frac{\boldsymbol{p}_j}{\|\boldsymbol{p}_j\|}, \sum_{j=1}^{n_b} \frac{\|\boldsymbol{p}_j\|\boldsymbol{u} - \frac{\langle \boldsymbol{u}, \boldsymbol{p}_j\rangle}{\|\boldsymbol{p}_j\|}\boldsymbol{p}_j}{\|\boldsymbol{p}_j\|^2} \Big\rangle$$

$$= 2 \Big\langle \sum_{j=1}^{n_b} \frac{\boldsymbol{p}_j}{\|\boldsymbol{p}_j\|}, \sum_{j=1}^{n_b} \frac{1}{\|\boldsymbol{p}_j\|}\Big( \boldsymbol{u} - \big\langle \boldsymbol{u}, \frac{\boldsymbol{p}_j}{\|\boldsymbol{p}_j\|}\big\rangle \frac{\boldsymbol{p}_j}{\|\boldsymbol{p}_j\|} \Big) \Big\rangle$$

$$= 2 \Big\langle \sum_{j=1}^{n_b} \boldsymbol{x}_j, \sum_{j=1}^{n_b} \frac{1}{\|\boldsymbol{p}_j\|}\Big( \boldsymbol{u} - \langle \boldsymbol{u}, \boldsymbol{x}_j\rangle \boldsymbol{x}_j \Big) \Big\rangle$$

$$= 2 \Big\langle -\boldsymbol{u}, \sum_{j=1}^{n_b} \frac{1}{\|\boldsymbol{p}_j\|}\Big( \boldsymbol{u} - \langle \boldsymbol{u}, \boldsymbol{x}_j\rangle \boldsymbol{x}_j \Big) \Big\rangle$$

$$= -2 \sum_{j=1}^{n_b} \frac{\|\boldsymbol{u}\|^2 - \langle \boldsymbol{u}, \boldsymbol{x}_j\rangle^2}{\|\boldsymbol{p}_j\|}$$

$$\leq -2 \sum_{j=1}^{n_b} \frac{\|\boldsymbol{u}\|^2 - \|\boldsymbol{u}\|^2\|\boldsymbol{x}_j\|^2}{\|\boldsymbol{p}_j\|}$$

$$= 0$$

Since the equality holds when $\langle \boldsymbol{u}, \boldsymbol{x}_j\rangle^2 = \|\boldsymbol{u}\|^2\|\boldsymbol{x}_j\|^2$ for all $j$, we have strict inequality when all $\boldsymbol{p}_i$'s are not located on a single ray from the origin. $\square$

## C.3 PROOF OF THEOREM 3

The proof of **Theorem 3** is very similar to that of **Lemma 2**.

**Theorem 3.** *Let $\boldsymbol{p}_1(\boldsymbol{w}_t^0), \boldsymbol{p}_2(\boldsymbol{w}_t^0), \cdots, \boldsymbol{p}_{n_b}(\boldsymbol{w}_t^0)$ be $d$-dimensional vectors, and all $\boldsymbol{p}_i(\boldsymbol{w}_t^0)$'s are not on a single ray from the current location $\boldsymbol{w}_t^0$. If*

$$\Big\| \sum_{i=1}^{n_b} \frac{\boldsymbol{p}_i(\boldsymbol{w}_t^0) - \boldsymbol{w}_t^0}{\|\boldsymbol{p}_i(\boldsymbol{w}_t^0) - \boldsymbol{w}_t^0\|} - \sum_{i=1}^{n_b} \frac{\hat{\boldsymbol{g}}_i(\boldsymbol{w}_t^{i-1})}{\|\hat{\boldsymbol{g}}_i(\boldsymbol{w}_t^{i-1})\|} \Big\| \leq \xi \tag{6}$$

*for sufficiently small $\xi > 0$, then there exists positive number $\eta$ such that*

$$\Big\| \sum_{j=1}^{n_b} \frac{\boldsymbol{p}_j(\boldsymbol{w}_t^0) - \boldsymbol{w}_t^0 - \epsilon \sum_{i=1}^{n_b} \frac{\hat{\boldsymbol{g}}_i(\boldsymbol{w}_t^{i-1})}{\|\hat{\boldsymbol{g}}_i(\boldsymbol{w}_t^{i-1})\|}}{\|\boldsymbol{p}_j(\boldsymbol{w}_t^0) - \boldsymbol{w}_t^0 - \epsilon \sum_{i=1}^{n_b} \frac{\hat{\boldsymbol{g}}_i(\boldsymbol{w}_t^{i-1})}{\|\hat{\boldsymbol{g}}_i(\boldsymbol{w}_t^{i-1})\|}\|} \Big\| < \Big\| \sum_{i=1}^{n_b} \frac{\boldsymbol{p}_i(\boldsymbol{w}_t^0) - \boldsymbol{w}_t^0}{\|\boldsymbol{p}_i(\boldsymbol{w}_t^0) - \boldsymbol{w}_t^0\|} \Big\| \tag{7}$$

*for all $\epsilon \in (0, \eta]$.*

*Proof.* We regard $\boldsymbol{w}_t^0$ as the origin $\boldsymbol{0}$. For simplicity, write $\boldsymbol{p}_i(0)$ and $\hat{\boldsymbol{g}}_i(\boldsymbol{w}_t^{i-1})$ as $\boldsymbol{p}_i$ and $\hat{\boldsymbol{g}}_i$, respectively. Let $f(\epsilon) = \left\| \sum_{j=1}^{n_b} \frac{\boldsymbol{p}_j - \epsilon \sum_{i=1}^{n_b} \frac{\boldsymbol{p}_i}{\|\boldsymbol{p}_i\|}}{\|\boldsymbol{p}_j - \epsilon \sum_{i=1}^{n_b} \frac{\boldsymbol{p}_i}{\|\boldsymbol{p}_i\|}\|} \right\|^2$ and $\tilde{f}(\epsilon) = \left\| \sum_{j=1}^{n_b} \frac{\boldsymbol{p}_j - \epsilon \sum_{i=1}^{n_b} \frac{\hat{\boldsymbol{g}}_i}{\|\hat{\boldsymbol{g}}_i\|}}{\|\boldsymbol{p}_j - \epsilon \sum_{i=1}^{n_b} \frac{\hat{\boldsymbol{g}}_i}{\|\hat{\boldsymbol{g}}_i\|}\|} \right\|^2$. Denote $\boldsymbol{u} = -\sum_{j=1}^{n_b} \frac{\boldsymbol{p}_i}{\|\boldsymbol{p}_i\|}$, $\boldsymbol{t} = \sum_{i=1}^{n_b} \frac{\boldsymbol{p}_i}{\|\boldsymbol{p}_i\|} - \sum_{i=1}^{n_b} \frac{\hat{\boldsymbol{g}}_i}{\|\hat{\boldsymbol{g}}_i\|}$ and $\tilde{\boldsymbol{p}}_j(\epsilon) = \boldsymbol{p}_j + \epsilon(\boldsymbol{u} + \boldsymbol{t})$. Then

$$\tilde{f}(\epsilon) = \left\| \sum_{i=1}^{n_b} \frac{\tilde{\boldsymbol{p}}_j(\epsilon)}{\|\tilde{\boldsymbol{p}}_j(\epsilon)\|} \right\|^2.$$

Now we differentiate $\tilde{f}(\epsilon)$ with respect to $\epsilon$, that is,

$$\tilde{f}'(\epsilon) = 2\Big\langle \sum_{j=1}^{n_b} \frac{\tilde{\boldsymbol{p}}_j(\epsilon)}{\|\tilde{\boldsymbol{p}}_j(\epsilon)\|}, \sum_{j=1}^{n_b} \frac{\|\tilde{\boldsymbol{p}}_j(\epsilon)\|(\boldsymbol{u} + \boldsymbol{t}) - \frac{\langle \boldsymbol{u}+\boldsymbol{t}, \tilde{\boldsymbol{p}}_j(\epsilon)\rangle}{\|\tilde{\boldsymbol{p}}_j(\epsilon)\|} \tilde{\boldsymbol{p}}_j(\epsilon)}{\|\tilde{\boldsymbol{p}}_j(\epsilon)\|^2} \Big\rangle.$$

Recall that $\tilde{\boldsymbol{p}}_j(0) = \boldsymbol{p}_j$. Rewrite $\frac{\boldsymbol{p}_j}{\|\boldsymbol{p}_j\|} = \boldsymbol{x}_j$ and use $f'(0)$ in the proof of **Lemma 2**

$$
\begin{aligned}
\tilde{f}'(0) &= 2\Big\langle \sum_{j=1}^{n_b} \frac{\boldsymbol{p}_j}{\|\boldsymbol{p}_j\|}, \sum_{j=1}^{n_b} \frac{\|\boldsymbol{p}_j\|(\boldsymbol{u} + \boldsymbol{t}) - \frac{\langle \boldsymbol{u}+\boldsymbol{t}, \boldsymbol{p}_j\rangle}{\|\boldsymbol{p}_j\|}\boldsymbol{p}_j}{\|\boldsymbol{p}_j\|^2} \Big\rangle \\
&= 2\Big\langle -\boldsymbol{u}, \sum_{j=1}^{n_b} \frac{\boldsymbol{u} + \boldsymbol{t} - \big\langle \boldsymbol{u}+\boldsymbol{t}, \frac{\boldsymbol{p}_j}{\|\boldsymbol{p}_j\|}\big\rangle \frac{\boldsymbol{p}_j}{\|\boldsymbol{p}_j\|}}{\|\boldsymbol{p}_j\|} \Big\rangle \\
&= 2\Big\langle -\boldsymbol{u}, \sum_{j=1}^{n_b} \frac{1}{\|\boldsymbol{p}_j\|}\big(\boldsymbol{u} + \boldsymbol{t} - \langle \boldsymbol{u}+\boldsymbol{t}, \boldsymbol{x}_j\rangle \boldsymbol{x}_j\big) \Big\rangle \\
&= 2\Big\langle -\boldsymbol{u}, \sum_{j=1}^{n_b} \frac{1}{\|\boldsymbol{p}_j\|}\big(\boldsymbol{u} - \langle \boldsymbol{u}, \boldsymbol{x}_j\rangle \boldsymbol{x}_j\big) \Big\rangle + 2\Big\langle -\boldsymbol{u}, \sum_{j=1}^{n_b} \frac{1}{\|\boldsymbol{p}_j\|}\big(\boldsymbol{t} - \langle \boldsymbol{t}, \boldsymbol{x}_j\rangle \boldsymbol{x}_j\big) \Big\rangle \\
&= f'(0) - 2\sum_{j=1}^{n_b} \frac{1}{\|\boldsymbol{p}_j\|}\big(\langle \boldsymbol{u}, \boldsymbol{t}\rangle - \langle \boldsymbol{t}, \boldsymbol{x}_j\rangle \langle \boldsymbol{u}, \boldsymbol{x}_j\rangle\big)
\end{aligned}
$$

Since $f'(0) < 0$ by the proof of **Lemma 2**,

$$\tilde{f}'(0) < 0 \quad \Longleftrightarrow \quad 2\sum_{j=1}^{n_b} \frac{1}{\|\boldsymbol{p}_j\|}\big(\langle \boldsymbol{t}, \boldsymbol{x}_j\rangle \langle \boldsymbol{u}, \boldsymbol{x}_j\rangle - \langle \boldsymbol{u}, \boldsymbol{t}\rangle\big) < |f'(0)|.$$

By using $\|\boldsymbol{x}_j\| = 1$ and applying the Cauchy inequality,

$$
\begin{aligned}
2\sum_{j=1}^{n_b} \frac{1}{\|\boldsymbol{p}_j\|}\big(\langle \boldsymbol{t}, \boldsymbol{x}_j\rangle \langle \boldsymbol{u}, \boldsymbol{x}_j\rangle - \langle \boldsymbol{u}, \boldsymbol{t}\rangle\big) &\leq 2\sum_{j=1}^{n_b} \frac{\|\boldsymbol{t}\|\|\boldsymbol{x}_j\|\|\boldsymbol{u}\|\|\boldsymbol{x}_j\| + \|\boldsymbol{u}\|\|\boldsymbol{t}\|}{\|\boldsymbol{p}_j\|} \\
&= 4\sum_{j=1}^{n_b} \frac{\|\boldsymbol{u}\|\|\boldsymbol{t}\|}{\|\boldsymbol{p}_j\|} \\
&\leq \frac{4n_b\|\boldsymbol{u}\|\|\boldsymbol{t}\|}{\min_j \|\boldsymbol{p}_j\|} \\
&\leq \frac{4n_b^2 \xi}{\min_j \|\boldsymbol{p}_j\|} \quad \Big(\because \|\boldsymbol{u}\| \leq \sum_{i=1}^{n_b} \|\boldsymbol{x}_i\| = n_b\Big).
\end{aligned}
$$

Define $r = \min_j \|\boldsymbol{p}_j\|$. If

$$\xi < \frac{|f'(0)|r}{4n_b^2},$$

then

$$\tilde{f}(\epsilon) < 0.$$

$\square$

# D    PROOFS FOR COROLLARY 3.1

**Corollary 3.1.** *Let $\boldsymbol{p}_i$ be local minibatch solutions for each $f_{\mathbb{I}_i}$. Suppose a region $\mathcal{R}$ satisfying:*

$$\text{For all } \boldsymbol{w}, \boldsymbol{w}' \in \mathcal{R}, \quad \boldsymbol{p}_i(\boldsymbol{w}) = \boldsymbol{p}_i(\boldsymbol{w}') = \boldsymbol{p}_i$$

*for all $i = 1, \cdots, n_b$. Further, assume that Hessian matrices of $f_{\mathbb{I}_i}$'s are positive definite, well-conditioned, and bounded in the sense of matrix $L^2$-norm on $\mathcal{R}$. If SGD moves $\boldsymbol{w}_t^0$ to $\boldsymbol{w}_{t+1}^0$ on $\mathcal{R}$ with a large batch size and a small learning rate, then $\hat{\kappa}(\boldsymbol{w}_t^0) > \hat{\kappa}(\boldsymbol{w}_{t+1}^0)$. Moreover, we can estimate $\hat{\kappa}(\boldsymbol{w}_t^0)$ and $\hat{\kappa}(\boldsymbol{w}_{t+1}^0)$ by minibatch gradients at $\boldsymbol{w}_t^0$ and $\boldsymbol{w}_{t+1}^0$, respectively.*

*Proof.* Recall that $\boldsymbol{w}_{t+1}^0 = \boldsymbol{w}_t^0 + \eta \sum_{i=1}^{n_b} \hat{\boldsymbol{g}}_i(\boldsymbol{w}_t^{i-1})$ where $\eta$ is a learning rate. To prove **Corollary 3.1**, we need to show $\hat{\kappa}(\boldsymbol{w}_{t+1}^0) < \hat{\kappa}(\boldsymbol{w}_t^0)$ which is equivalent to

$$\left\| \sum_{j=1}^{n_b} \frac{\boldsymbol{p}_j(\boldsymbol{w}_{t+1}^0) - \boldsymbol{w}_t^0 - \eta \sum_{i=1}^{n_b} \hat{\boldsymbol{g}}_i(\boldsymbol{w}_t^{i-1})}{\|\boldsymbol{p}_j(\boldsymbol{w}_{t+1}^0) - \boldsymbol{w}_t^0 - \eta \sum_{i=1}^{n_b} \hat{\boldsymbol{g}}_i(\boldsymbol{w}_t^{i-1})\|} \right\| < \left\| \sum_{i=1}^{n_b} \frac{\boldsymbol{p}_i(\boldsymbol{w}_t^0) - \boldsymbol{w}_t^0}{\|\boldsymbol{p}_i(\boldsymbol{w}_t^0) - \boldsymbol{w}_t^0\|} \right\|. \tag{8}$$

Since $\|\nabla_{\boldsymbol{w}}^2 f_{\mathbb{I}_i}(\cdot)\|_2$ is bounded on $\mathcal{R}$, $\nabla_{\boldsymbol{w}} f_{\mathbb{I}_i}(\cdot)$ is Lipschitz continuous on $\mathcal{R}$(Bottou, 2010). If the batch size is sufficiently large and the learning rate $\eta$ is sufficiently small, $\|\hat{\boldsymbol{g}}_i(\boldsymbol{w}_t^{i-1})\| \approx \|\hat{\boldsymbol{g}}_i(\boldsymbol{w}_t^0)\| \approx \tau$ for all $i$ by **Theorem 1**. Therefore, we have

$$\eta \sum_{i=1}^{n_b} \hat{\boldsymbol{g}}_i(\boldsymbol{w}_t^{i-1}) \approx \tau \eta \sum_{i=1}^{n_b} \frac{\hat{\boldsymbol{g}}_i(\boldsymbol{w}_t^{i-1})}{\|\hat{\boldsymbol{g}}_i(\boldsymbol{w}_t^{i-1})\|}$$

If we denote $\tau\eta$ as $\epsilon$, we can convert (8) to (9).

$$\left\| \sum_{j=1}^{n_b} \frac{\boldsymbol{p}_j(\boldsymbol{w}_{t+1}^0) - \boldsymbol{w}_t^0 - \epsilon \sum_{i=1}^{n_b} \frac{\hat{\boldsymbol{g}}_i(\boldsymbol{w}_t^{i-1})}{\|\hat{\boldsymbol{g}}_i(\boldsymbol{w}_t^{i-1})\|}}{\|\boldsymbol{p}_j(\boldsymbol{w}_{t+1}^0) - \boldsymbol{w}_t^0 - \epsilon \sum_{i=1}^{n_b} \frac{\hat{\boldsymbol{g}}_i(\boldsymbol{w}_t^{i-1})}{\|\hat{\boldsymbol{g}}_i(\boldsymbol{w}_t^{i-1})\|}\|} \right\| < \left\| \sum_{i=1}^{n_b} \frac{\boldsymbol{p}_i(\boldsymbol{w}_t^0) - \boldsymbol{w}_t^0}{\|\boldsymbol{p}_i(\boldsymbol{w}_t^0) - \boldsymbol{w}_t^0\|} \right\|. \tag{9}$$

Since both $\boldsymbol{w}_{t+1}^0$ and $\boldsymbol{w}_t^0$ are in $\mathcal{R}$ for small learning rate, we have $\boldsymbol{p}_i(\boldsymbol{w}_{t+1}^0) = \boldsymbol{p}_i(\boldsymbol{w}_t^0) = \boldsymbol{p}_i$ by the assumption. That is, (9) is equivalent to (7). In (7), $\hat{\boldsymbol{g}}_i(\boldsymbol{w}_t^{i-1})/\|\hat{\boldsymbol{g}}_i(\boldsymbol{w}_t^{i-1})\|$ cannot be replaced by $(\boldsymbol{p}_i - \boldsymbol{w}_t^0)/\|\boldsymbol{p}_i - \boldsymbol{w}_t^0\|$ in general. Hence we introduce **Definition D.1** and **Lemma D.1** to connect the direction of the minibatch gradient with the corresponding local minibatch solution.

**Definition D.1.** *The condition number $c(\boldsymbol{A})$ of a matrix $\boldsymbol{A}$ is defined as*

$$c(\boldsymbol{A}) = \frac{\sigma_{\max}(\boldsymbol{A})}{\sigma_{\min}(\boldsymbol{A})}$$

*where $\sigma_{\max}(\boldsymbol{A})$ and $\sigma_{\min}(\boldsymbol{A})$ are maximal and minimal singular values of $\boldsymbol{A}$, respectively. If $\boldsymbol{A}$ is positive-definite matrix, then*

$$c(\boldsymbol{A}) = \frac{\lambda_{\max}(\boldsymbol{A})}{\lambda_{\min}(\boldsymbol{A})}.$$

*Here $\lambda_{\max}(\boldsymbol{A})$ and $\lambda_{\min}(\boldsymbol{A})$ are maximal and minimal eigenvalues of $\boldsymbol{A}$, respectively.*

**Lemma D.1.** *If the condition number of the positive definite Hessian matrix of $f_{\mathbb{I}_i}$ at a local minibatch solution $\boldsymbol{p}_i$, denoted by $\boldsymbol{H}_i = \nabla_{\boldsymbol{w}}^2 f_{\mathbb{I}_i}(\boldsymbol{p}_i)$, is close to 1 (well-conditioned), then the direction to $\boldsymbol{p}_i$ from $\boldsymbol{w}$ is approximately parallel to its negative gradient at $\boldsymbol{w}$. That is, for all $\boldsymbol{w} \in \mathcal{R}$,*

$$\left\| \frac{\boldsymbol{p}_i - \boldsymbol{w}}{\|\boldsymbol{p}_i - \boldsymbol{w}\|} - \frac{\hat{\boldsymbol{g}}_i(\boldsymbol{w})}{\|\hat{\boldsymbol{g}}_i(\boldsymbol{w})\|} \right\| \approx 0$$

*where $\hat{\boldsymbol{g}}_i(\boldsymbol{w}) = -\nabla_{\boldsymbol{w}} f_{\mathbb{I}_i}(\boldsymbol{w})$.*

*Proof.* By the second order Taylor expansion,

$$f_{\mathbb{I}_i}(\boldsymbol{w}) \approx f_{\mathbb{I}_i}(\boldsymbol{p}_i) + \frac{1}{2}(\boldsymbol{w} - \boldsymbol{p}_i)^\top \boldsymbol{H}_i(\boldsymbol{w} - \boldsymbol{p}_i).$$

Hence,

$$\hat{g}_i(w) = -\nabla_w f_{\mathbb{I}_i}(w) \approx -H_i(w - p_i)$$

Denote $p_i - w$ as $x$. Then, we only need to show

$$\left\| \frac{x}{\|x\|} - \frac{H_i x}{\|H_i x\|} \right\|^2 \approx 0$$

Since $H_i$ is positive definite, we can diagonalize it as $H_i = P_i^\top \Lambda_i P_i$ where $P_i$ is an orthonormal transition matrix for $H_i$.

$$
\begin{aligned}
\left\| \frac{x}{\|x\|} - \frac{H_i x}{\|H_i x\|} \right\|^2 &= 2 - 2 \frac{x^\top H_i x}{\|x\| \|H_i x\|} \\
&= 2 - 2 \frac{(P_i x)^\top \Lambda_i P_i x}{\|P_i x\| \|P_i^\top \Lambda_i P_i x\|} \\
&= 2 - 2 \frac{(P_i x)^\top \Lambda_i P_i x}{\|P_i x\| \|\Lambda_i P x\|} \\
&\leq 2 - 2 \frac{\sum_j \lambda_j (P_i x)_j^2}{\|P_i x\| \|\Lambda_i P_i x\|} \\
&\leq 2 - 2 \frac{\lambda_{\min} \|P_i x\|^2}{\|P_i x\| \lambda_{\max} \|P_i x\|} \\
&= 2 - 2 \frac{\lambda_{\min}}{\lambda_{\max}} \approx 0
\end{aligned}
$$

$\square$

**Lemma D.1** suggests that a well-conditioned Hessian matrix of $f_{\mathbb{I}_i}$ at $p_i$ allows $\hat{g}_i(w)/\|\hat{g}_i(w)\|$ to be replaced by $(p_i - w)/\|p_i - w\|$ for all $w \in \mathcal{R}$. Using this, we prove **Lemma D.2**.

**Lemma D.2.** *Let $w$ be a parameter in $\mathcal{R}$. If the condition number of Hessian matrix of $f_{\mathbb{I}_i}$ is sufficiently close to 1 (well-conditioned) and $\frac{\|w - w_t^0\|}{\|p_i - w_t^0\|}$ is sufficiently close to 0, then*

$$\left\| \frac{p_i - w_t^0}{\|p_i - w_t^0\|} - \frac{\hat{g}_i(w)}{\|\hat{g}_i(w)\|} \right\| \leq \frac{\xi}{n_b}$$

*for all sufficiently small $\xi$.*

*Proof.* We have

$$
\begin{aligned}
\left\| \frac{p_i - w_t^0}{\|p_i - w_t^0\|} - \frac{\hat{g}_i(w)}{\|\hat{g}_i(w)\|} \right\| &\leq \left\| \frac{p_i - w_t^0}{\|p_i - w_t^0\|} - \frac{\tilde{g}(w)}{\|\tilde{g}(w)\|} \right\| + \left\| \frac{\tilde{g}(w)}{\|\tilde{g}(w)\|} - \frac{\hat{g}_i(w)}{\|\hat{g}_i(w)\|} \right\| \\
&\leq \left\| \frac{p_i - w_t^0}{\|p_i - w_t^0\|} - \frac{\tilde{g}(w)}{\|\tilde{g}(w)\|} \right\| + \left\| \frac{\tilde{g}(w)}{\|\tilde{g}(w)\|} - \frac{p_i - w_t^0}{\|p_i - w_t^0\|} \right\| \\
&\quad + \left\| \frac{p_i - w_t^0}{\|p_i - w_t^0\|} - \frac{p_i - w}{\|p_i - w\|} \right\| + \left\| \frac{p_i - w}{\|p_i - w\|} - \frac{\hat{g}_i(w)}{\|\hat{g}_i(w)\|} \right\| \\
&\leq \epsilon + \epsilon + \sqrt{2\left(1 - \frac{\langle p_i - w_t^0, p_i - w \rangle}{\|p_i - w_t^0\| \|p_i - w\|}\right)} + \epsilon \\
&= 3\epsilon + \sqrt{2\left(1 - \frac{\|p_i - w_t^0\|^2 - \langle p_i - w_t^0, w - w_t^0 \rangle}{\|p_i - w_t^0\| \|p_i - w\|}\right)}
\end{aligned}
$$

for sufficently small $\epsilon$(See **Lemma D.1**). Now we only need to show

$$\sqrt{2\left(1 - \frac{\|p_i - w_t^0\|^2 - \langle p_i - w_t^0, w - w_t^0 \rangle}{\|p_i - w_t^0\| \|p_i - w\|}\right)} < \epsilon \,. \tag{10}$$

Since $\frac{\|\boldsymbol{w}-\boldsymbol{w}_t^0\|}{\|\boldsymbol{p}_i-\boldsymbol{w}_t^0\|}$ is sufficiently small, we have

$$
\begin{aligned}
\|\boldsymbol{p}_i - \boldsymbol{w}_t^0\|^2 - \langle \boldsymbol{p}_i - \boldsymbol{w}_t^0, \boldsymbol{w} - \boldsymbol{w}_t^0 \rangle &= \|\boldsymbol{p}_i - \boldsymbol{w}_t^0\|\Big( \|\boldsymbol{p}_i - \boldsymbol{w}_t^0\| - \Big\langle \frac{\boldsymbol{p}_i - \boldsymbol{w}_t^0}{\|\boldsymbol{p}_i - \boldsymbol{w}_t^0\|}, \boldsymbol{w} - \boldsymbol{w}_t^0 \Big\rangle \Big) \\
&\geq \|\boldsymbol{p}_i - \boldsymbol{w}_t^0\|(\|\boldsymbol{p}_i - \boldsymbol{w}_t^0\| - \|\boldsymbol{w} - \boldsymbol{w}_t^0\|) \\
&= \|\boldsymbol{p}_i - \boldsymbol{w}_t^0\|^2 \Big( 1 - \frac{\|\boldsymbol{w} - \boldsymbol{w}_t^0\|}{\|\boldsymbol{p}_i - \boldsymbol{w}_t^0\|} \Big) \\
&\geq 0.
\end{aligned}
$$

By using the above non-negativeness, we have the following inequality.

$$
\begin{aligned}
1 &\geq \frac{\|\boldsymbol{p}_i - \boldsymbol{w}_t^0\|^2 - \langle \boldsymbol{p}_i - \boldsymbol{w}_t^0, \boldsymbol{w} - \boldsymbol{w}_t^0 \rangle}{\|\boldsymbol{p}_i - \boldsymbol{w}_t^0\|\|\boldsymbol{p}_i - \boldsymbol{w}\|} \\
&\geq \frac{\|\boldsymbol{p}_i - \boldsymbol{w}_t^0\|^2 - \langle \boldsymbol{p}_i - \boldsymbol{w}_t^0, \boldsymbol{w} - \boldsymbol{w}_t^0 \rangle}{\|\boldsymbol{p}_i - \boldsymbol{w}_t^0\|(\|\boldsymbol{p}_i - \boldsymbol{w}_t^0\| + \|\boldsymbol{w} - \boldsymbol{w}_t^0\|)} \\
&= \frac{\frac{\|\boldsymbol{p}_i - \boldsymbol{w}_t^0\|}{\|\boldsymbol{w} - \boldsymbol{w}_t^0\|} - \big\langle \frac{\boldsymbol{p}_i - \boldsymbol{w}_t^0}{\|\boldsymbol{p}_i - \boldsymbol{w}_t^0\|}, \frac{\boldsymbol{w} - \boldsymbol{w}_t^0}{\|\boldsymbol{w} - \boldsymbol{w}_t^0\|} \big\rangle}{1 + \frac{\|\boldsymbol{p}_i - \boldsymbol{w}_t^0\|}{\|\boldsymbol{w} - \boldsymbol{w}_t^0\|}} \\
&\geq \frac{\frac{\|\boldsymbol{p}_i - \boldsymbol{w}_t^0\|}{\|\boldsymbol{w} - \boldsymbol{w}_t^0\|} - 1}{1 + \frac{\|\boldsymbol{p}_i - \boldsymbol{w}_t^0\|}{\|\boldsymbol{w} - \boldsymbol{w}_t^0\|}}.
\end{aligned}
\tag{11}
$$

As $\frac{\|\boldsymbol{w} - \boldsymbol{w}_t^0\|}{\|\boldsymbol{p}_i - \boldsymbol{w}_t^0\|} \to 0^+$, (11) is monotonically increasing to 1. This implies that (10) holds for sufficiently small $\epsilon$. $\qquad\square$

With a small learning rate, $\boldsymbol{w}_t^{i-1}$'s are in $\mathcal{R}$ for all $i \in \{1, \ldots, n_b\}$. As a result, by **Lemma D.2**, we have

$$
\Big\| \frac{\boldsymbol{p}_i - \boldsymbol{w}_t^0}{\|\boldsymbol{p}_i - \boldsymbol{w}_t^0\|} - \frac{\hat{\boldsymbol{g}}_i(\boldsymbol{w}_t^{i-1})}{\|\hat{\boldsymbol{g}}_i(\boldsymbol{w}_t^{i-1})\|} \Big\| \leq \frac{\xi}{n_b}
\tag{12}
$$

for sufficiently small $\xi$. This implies (6) since

$$
\Big\| \sum_{i=1}^{n_b} \frac{\boldsymbol{p}_i - \boldsymbol{w}}{\|\boldsymbol{p}_i - \boldsymbol{w}\|} - \sum_{i=1}^{n_b} \frac{\hat{\boldsymbol{g}}_i(\boldsymbol{w}_t^{i-1})}{\|\hat{\boldsymbol{g}}_i(\boldsymbol{w}_t^{i-1})\|} \Big\| \leq \sum_{i=1}^{n_b} \Big\| \frac{\boldsymbol{p}_i - \boldsymbol{w}}{\|\boldsymbol{p}_i - \boldsymbol{w}\|} - \frac{\hat{\boldsymbol{g}}_i(\boldsymbol{w}_t^{i-1})}{\|\hat{\boldsymbol{g}}_i(\boldsymbol{w}_t^{i-1})\|} \Big\|.
$$

Then we can apply **Theorem 3** and $\hat{\kappa}(\boldsymbol{w}_t^0) > \hat{\kappa}(\boldsymbol{w}_{t+1}^0)$ holds.

For the last statement, *"Moreover, we can estimate $\hat{\kappa}(\boldsymbol{w}_t^0)$ and $\hat{\kappa}(\boldsymbol{w}_{t+1}^0)$ by minibatch gradients at $\boldsymbol{w}_t^0$ and $\boldsymbol{w}_{t+1}^0$, respectively."*, recall that

$$
\hat{\kappa}(\boldsymbol{w}_t^0) = h\Big( \Big\| \sum_{i=1}^{n_b} \frac{\boldsymbol{p}_i(\boldsymbol{w}_t^0) - \boldsymbol{w}_t^0}{\|\boldsymbol{p}_i(\boldsymbol{w}_t^0) - \boldsymbol{w}_t^0\|} \Big\| \Big)
$$

where $h(\cdot)$ is increasing and Lipschitz continuous(**Lemma 1**). By **Lemma D.1**, we have

$$
\Big\| \frac{\boldsymbol{p}_i(\boldsymbol{w}_t^0) - \boldsymbol{w}_t^0}{\|\boldsymbol{p}_i(\boldsymbol{w}_t^0) - \boldsymbol{w}_t^0\|} - \frac{\hat{\boldsymbol{g}}_i(\boldsymbol{w}_t^0)}{\|\hat{\boldsymbol{g}}_i(\boldsymbol{w}_t^0)\|} \Big\| < \frac{\xi}{n_b}
$$

for sufficiently small $\xi > 0$. Therefore,

$$
\Big| \Big\| \sum_{i=1}^{n_b} \frac{\boldsymbol{p}_i(\boldsymbol{w}_t^0) - \boldsymbol{w}_t^0}{\|\boldsymbol{p}_i(\boldsymbol{w}_t^0) - \boldsymbol{w}_t^0\|} \Big\| - \Big\| \sum_{i=1}^{n_b} \frac{\hat{\boldsymbol{g}}_i(\boldsymbol{w}_t^0)}{\|\hat{\boldsymbol{g}}_i(\boldsymbol{w}_t^0)\|} \Big\| \Big| \leq \Big\| \sum_{i=1}^{n_b} \frac{\boldsymbol{p}_i(\boldsymbol{w}_t^0) - \boldsymbol{w}_t^0}{\|\boldsymbol{p}_i(\boldsymbol{w}_t^0) - \boldsymbol{w}_t^0\|} - \sum_{i=1}^{n_b} \frac{\hat{\boldsymbol{g}}_i(\boldsymbol{w}_t^0)}{\|\hat{\boldsymbol{g}}_i(\boldsymbol{w}_t^0)\|} \Big\|
$$

where rhs is bounded by $\xi$. Hence, Lipschitz continuity of $h(\cdot)$ implies that

$$
\Big| h\Big( \Big\| \sum_{i=1}^{n_b} \frac{\boldsymbol{p}_i(\boldsymbol{w}_t^0) - \boldsymbol{w}_t^0}{\|\boldsymbol{p}_i(\boldsymbol{w}_t^0) - \boldsymbol{w}_t^0\|} \Big\| \Big) - h\Big( \Big\| \sum_{i=1}^{n_b} \frac{\hat{\boldsymbol{g}}_i(\boldsymbol{w}_t^0)}{\|\hat{\boldsymbol{g}}_i(\boldsymbol{w}_t^0)\|} \Big\| \Big) \Big| \to 0
$$

as $\xi \to 0$. That is,

$$
\hat{\kappa}(\boldsymbol{w}_t^0) \approx h\Big( \Big\| \sum_{i=1}^{n_b} \frac{\hat{\boldsymbol{g}}_i(\boldsymbol{w}_t^0)}{\|\hat{\boldsymbol{g}}_i(\boldsymbol{w}_t^0)\|} \Big\| \Big).
$$

Since $t$ is arbitrary, we can apply this for all $\boldsymbol{w} \in \mathcal{R}$ including $\boldsymbol{w}_{t+1}^0$. $\qquad\square$

# E EXPERIMENTAL DETAILS

## E.1 MODEL ARCHITECTURE

For all cases, their weighted layers do not have biases, and dropout (Srivastava et al., 2014) is not applied. We use Xavier initializations(Glorot & Bengio, 2010) and cross entropy loss functions for all experiments.

**FNN** The **FNN** is a fully connected network with a single hidden layer. It has 800 hidden units with ReLU (Nair & Hinton, 2010) activations and a softmax output layer.

**DFNN** The **DFNN** is a fully connected network with three hidden layers. It has 800 hidden units with ReLU activations in each hidden layers and a softmax output layer.

**CNN** The network architecture of **CNN** is similar to the network introduced in He et al. (2016) as a CIFAR-10 plain network. The first layer is $3 \times 3$ convolution layer and the number of output filters are 16. After that, we stack of $\{4, 4, 3, 1\}$ layers with $3 \times 3$ convolutions on the feature maps of sizes $\{32, 16, 8, 4\}$ and the numbers of filters $\{16, 32, 64, 128\}$, respectively. The subsampling is performed with a stride of 2. All convolution layers are activated by ReLU and the convolution part ends with a global average pooling(Lin et al., 2013), a 10-way fully-conneted layers, and softmax. Note that there are 14 stacked weighted layers.

**+BN** We apply batch normalization right before the ReLU activations on all hidden layers.

**+Res** The identity skip connections are added after every two convolution layers before ReLU nonlinearity (After batch normalization, if it is applied on it.). We concatenate zero padding slices backwards when the number of filters increases.

## E.2 DATA

We use neither data augmentations nor preprocessings except scaling pixel values into $[0, 1]$ both MNIST and CIFAR-10. In the case of CIFAR-10, for validation, we randomly choose 5000 images out of 50000 training images.

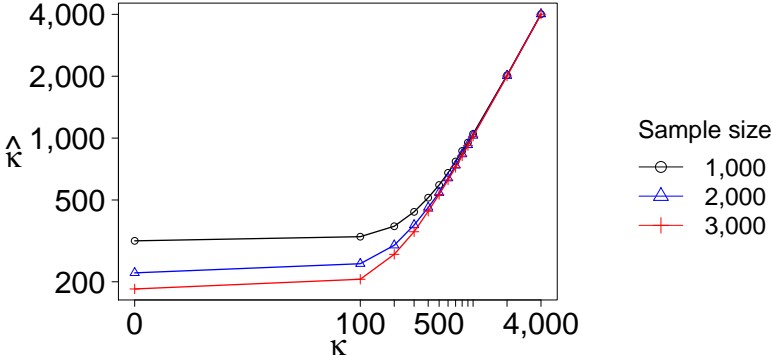

Figure 8: We show $\hat{\kappa}$ estimated from $\{1,000$ (black), $2,000$ (blue), $3,000$ (red)$\}$ random samples of the vMF distribution with underlying true $\kappa$ in $10,000$-dimensional space, as the function of $\kappa$ (in log-log scale except 0). For large $\kappa$, it is well-estimated by $\hat{\kappa}$ regardless of sample sizes. When the true $\kappa$ approaches 0, we need a larger sample size to more accurately estimate this.

Table 1: The value of $\hat{\kappa}$ (mean$\pm$std.) estimated from $\{1,000,\ 2,000,\ 3,000\}$ random samples of the vMF distribution with $\kappa = 0$ across various dimensions.

| Network | Dimension | $\hat{\kappa}$ (1,000 samples) | $\hat{\kappa}$ (2,000 samples) | $\hat{\kappa}$ (3,000 samples) |
|---|---|---|---|---|
| **CNN** | $206, 128$ | $6,527.09 \pm 5.98$ | $4,608.16 \pm 6.18$ | $3,762.45 \pm 7.23$ |
| **CNN+Res** | $206, 128$ | $6,527.09 \pm 5.98$ | $4,608.16 \pm 6.18$ | $3,762.45 \pm 7.23$ |
| **CNN+BN** | $207, 152$ | $6,563.62 \pm 8.68$ | $4,633.84 \pm 5.60$ | $3,781.04 \pm 3.12$ |
| **CNN+Res+BN** | $207, 152$ | $6,563.62 \pm 8.68$ | $4,633.84 \pm 5.60$ | $3,781.04 \pm 3.12$ |
| **FNN** | $635, 200$ | $20,111.90 \pm 13.04$ | $14,196.89 \pm 14.91$ | $11,607.39 \pm 9.27$ |
| **FNN+BN** | $636, 800$ | $20,157.57 \pm 14.06$ | $14,259.83 \pm 16.38$ | $11,621.63 \pm 6.83$ |
| **DFNN** | $1,915, 200$ | $60,619.02 \pm 13.49$ | $42,849.86 \pm 18.90$ | $34,983.31 \pm 15.62$ |
| **DFNN+BN** | $1,920, 000$ | $60,789.84 \pm 17.93$ | $42,958.71 \pm 25.61$ | $35,075.99 \pm 12.39$ |

## F   SOME NOTES ABOUT THE $\kappa$ ESTIMATE

We point out that, for a small $\kappa$, the absolute value of $\hat{\kappa}$ is not a precise indicator of the uniformity due to its dependence on the dimensionality, as was investigated earlier by Cutting et al. (2017). In order to verify this claim, we run some simulations. First, we vary the number of samples and the true underlying $\kappa$ with the fixed dimensionality (Unfortunately, we could not easily go over $10,000$ dimensions due to the difficulty in sampling from the vMF distribution with positive $\kappa$.). We draw $\{1,000,\ 2,000,\ 3,000\}$ random samples from the vMF distribution with the designated $\kappa$. We compute $\hat{\kappa}$ from these samples.

As can be seen from Figure 8, the $\hat{\kappa}$ approaches the true $\kappa$ from above as the number of samples increases. When the true $\kappa$ is large, the estimation error rapidly becomes zero as the number of samples approaches $3,000$. When the true $\kappa$ is low, however, the gap does not narrow completely even with $3,000$ samples.

While fixing the true $\kappa$ to 0 and the number of samples to $\{1,000,\ 2,000,\ 3,000\}$, we vary the dimensionality to empirically investigate the $\hat{\kappa}$. We choose to use $3,000$ samples to be consistent with our experiments in this paper. We run five simulations each and report both mean and standard deviation (Table 1).

We clearly observe the trend of increasing $\hat{\kappa}$'s with respect to the dimensions. This suggests that we should not compare the absolute values of $\hat{\kappa}$'s across different network architectures due to the differences in the number of parameters. This agrees well with Cutting et al. (2017) which empirically showed that the threshold for rejecting the null hypothesis of $\kappa = p$ by using $\hat{\kappa}$ where $p$ is a fixed value grows with respect to the dimensions.

## G    OTHER FOUR TRAINING RUNS IN FIGURE 7

We show plots from other four training runs in Figure 6. For all runs, the curves of GS (inverse of SNR) and $\hat{\kappa}$ are strongly correlated while GNS (inverse of normSNR) is less correlated to GS.

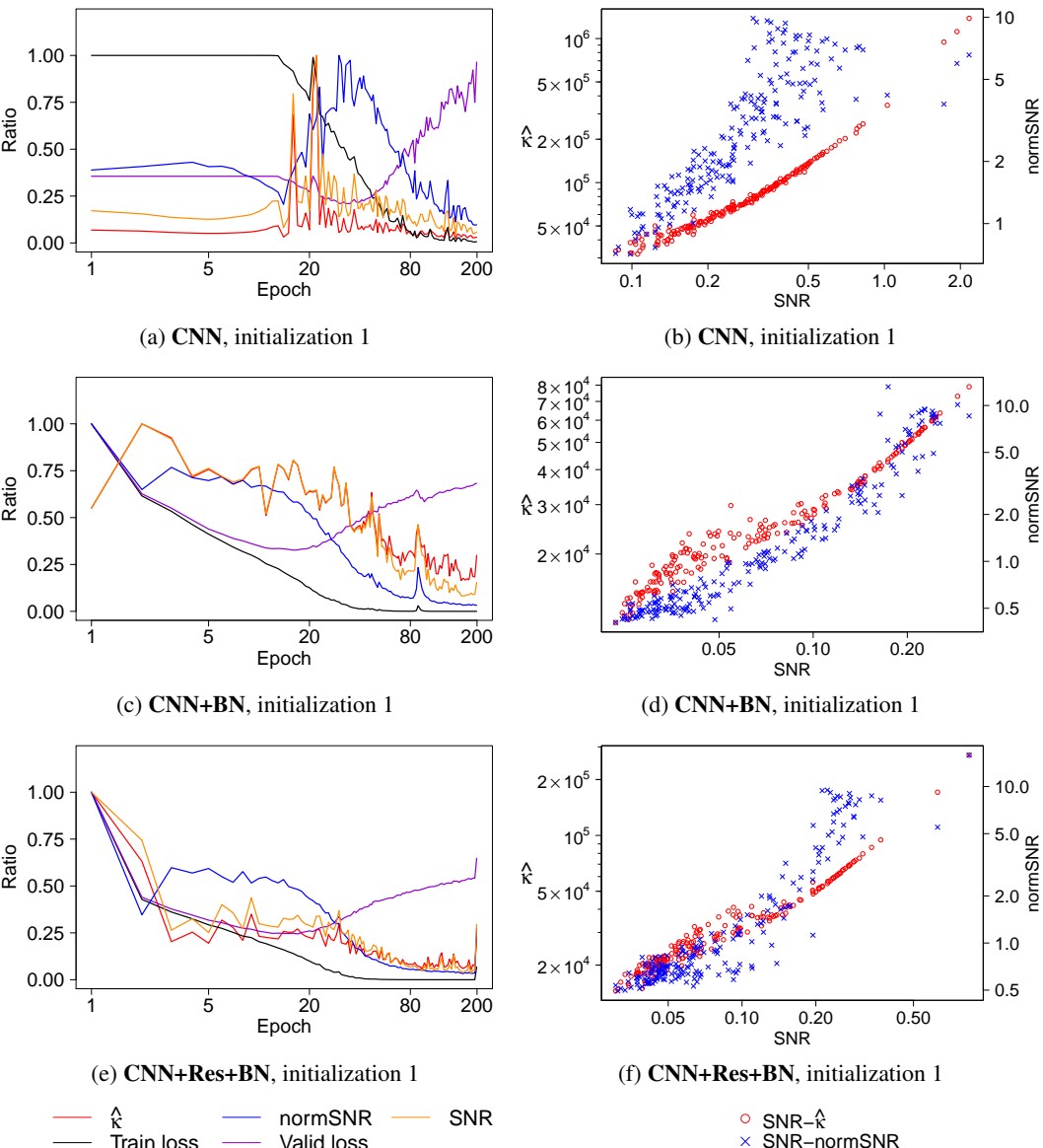

Figure 9: (a,c,e) We plot the evolution of the training loss (Train loss), validation loss (Valid loss), inverse of gradient stochasticity (SNR), inverse of gradient norm stochasticity (normSNR) and directional uniformity $\kappa$. We normalized each quantity by its maximum value over training for easier comparison on a single plot. In all the cases, SNR (orange) and $\hat{\kappa}$ (red) are almost entirely correlated with each other, while normSNR is less correlated. (b,d,f) We further verify this by illustrating SNR-$\hat{\kappa}$ scatter plots (red) and SNR-normSNR scatter plots (blue) in log-log scales. These plots suggest that the SNR is largely driven by the directional uniformity.

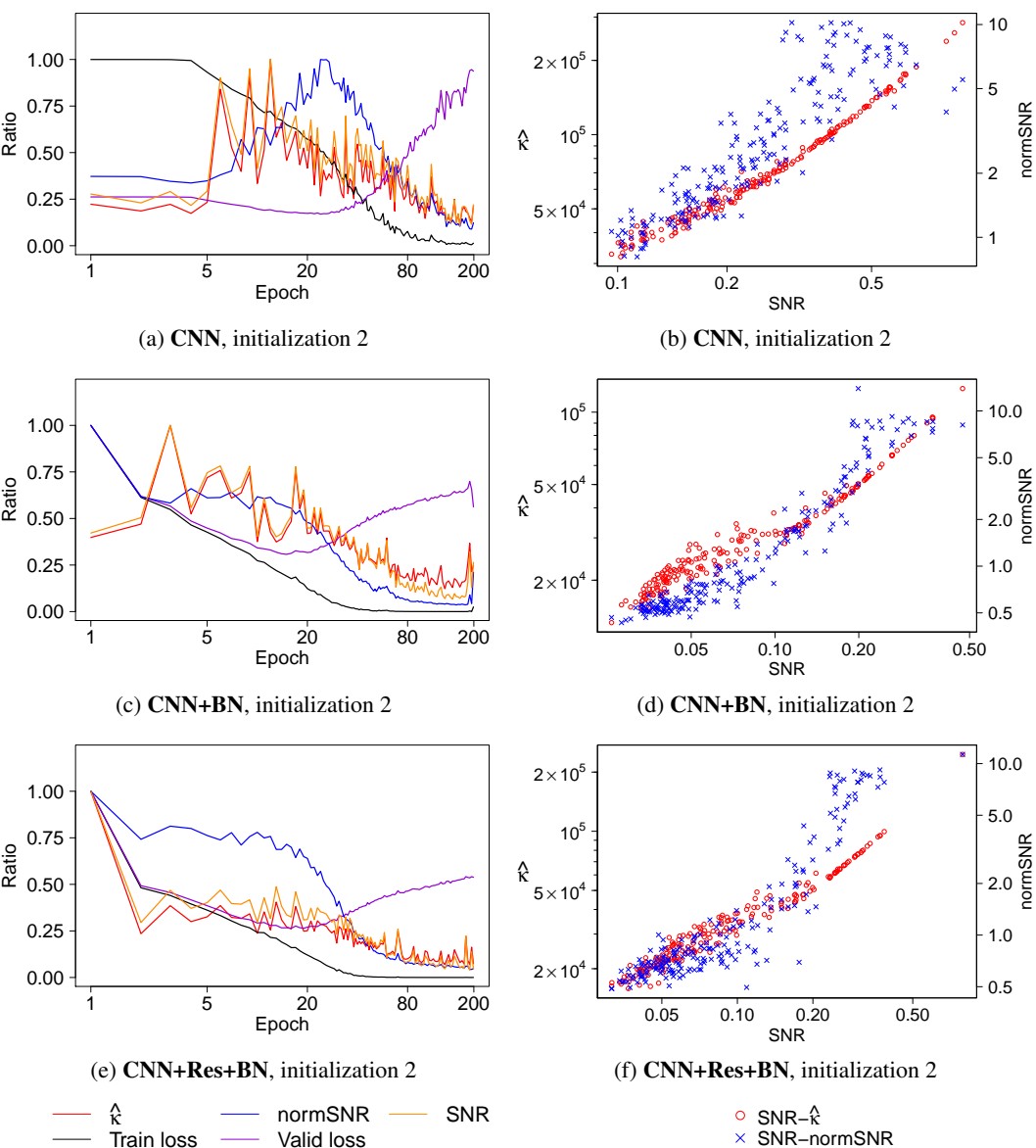

Figure 10: (a,c,e) We plot the evolution of the training loss (Train loss), validation loss (Valid loss), inverse of gradient stochasticity (SNR), inverse of gradient norm stochasticity (normSNR) and directional uniformity $\kappa$. We normalized each quantity by its maximum value over training for easier comparison on a single plot. In all the cases, SNR (orange) and $\hat{\kappa}$ (red) are almost entirely correlated with each other, while normSNR is less correlated. (b,d,f) We further verify this by illustrating SNR-$\hat{\kappa}$ scatter plots (red) and SNR-normSNR scatter plots (blue) in log-log scales. These plots suggest that the SNR is largely driven by the directional uniformity.

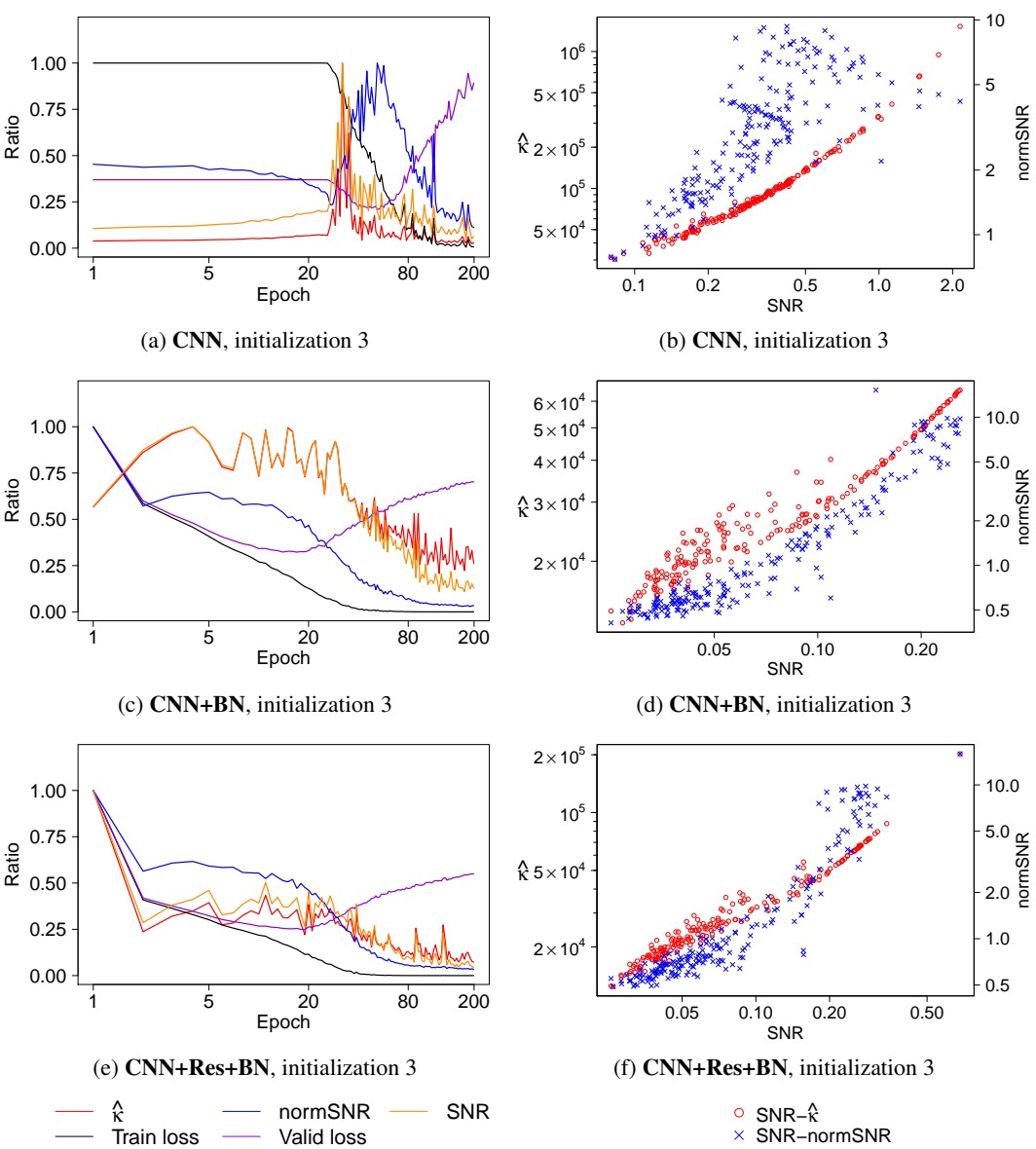

Figure 11: (a,c,e) We plot the evolution of the training loss (Train loss), validation loss (Valid loss), inverse of gradient stochasticity (SNR), inverse of gradient norm stochasticity (normSNR) and directional uniformity $\hat{\kappa}$. We normalized each quantity by its maximum value over training for easier comparison on a single plot. In all the cases, SNR (orange) and $\kappa$ (red) are almost entirely correlated with each other, while normSNR is less correlated. (b,d,f) We further verify this by illustrating SNR-$\hat{\kappa}$ scatter plots (red) and SNR-normSNR scatter plots (blue) in log-log scales. These plots suggest that the SNR is largely driven by the directional uniformity.

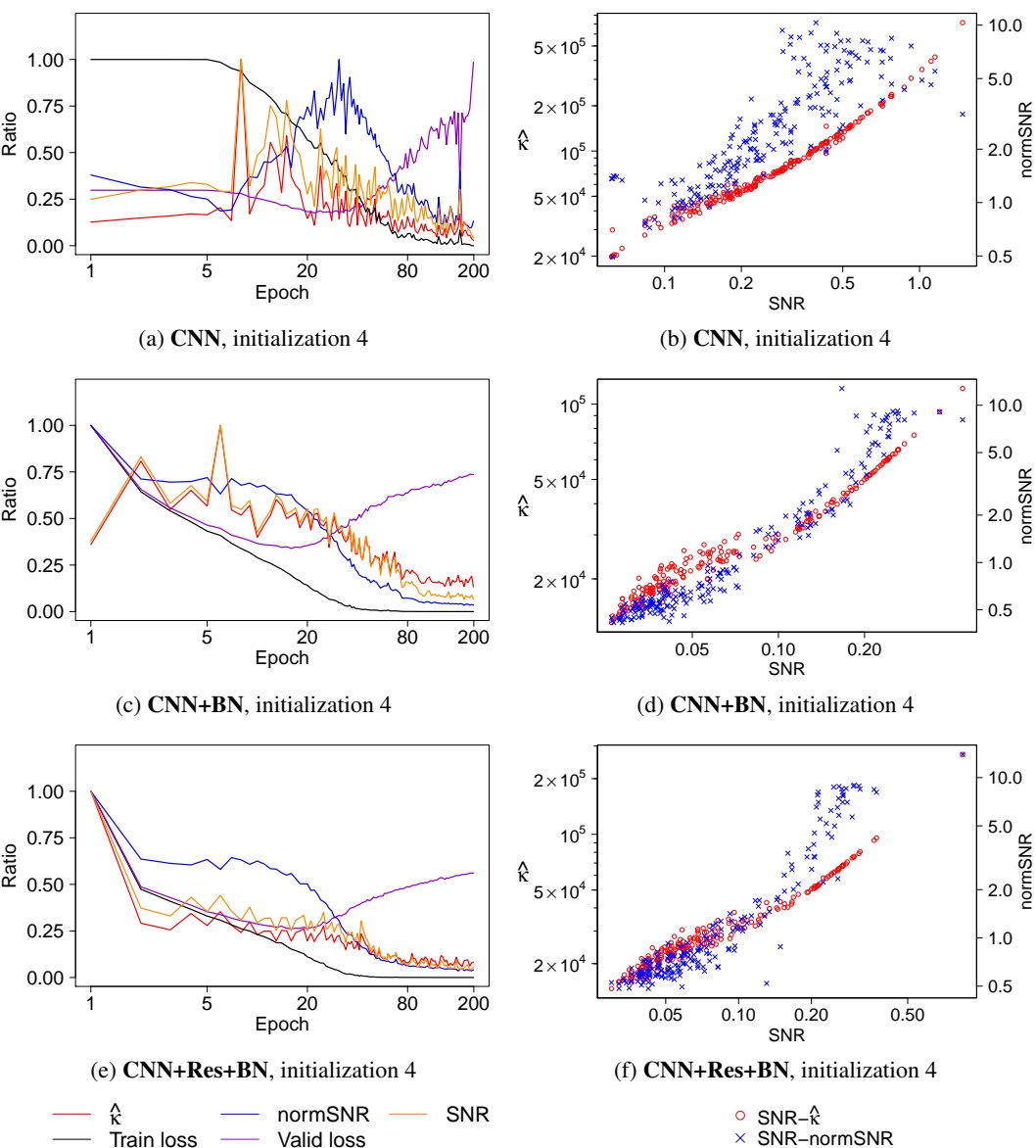

Figure 12: (a,c,e) We plot the evolution of the training loss (Train loss), validation loss (Valid loss), inverse of gradient stochasticity (SNR), inverse of gradient norm stochasticity (normSNR) and directional uniformity $\kappa$. We normalized each quantity by its maximum value over training for easier comparison on a single plot. In all the cases, SNR (orange) and $\hat{\kappa}$ (red) are almost entirely correlated with each other, while normSNR is less correlated. (b,d,f) We further verify this by illustrating SNR-$\hat{\kappa}$ scatter plots (red) and SNR-normSNR scatter plots (blue) in log-log scales. These plots suggest that the SNR is largely driven by the directional uniformity.

