# OpenReview forum: "Directional Analysis of Stochastic Gradient Descent via von Mises-Fisher Distributions in Deep Learning"
_ICLR.cc/2019/Conference_

### Official Review · AnonReviewer2 · 2018-11-02
**Contribution not entirely clear**

**Rating:** 4
**Confidence:** 3

**Review:**

Summary: This work provides an analysis of the directional distribution of of stochastic gradients in SGD. The basic claim is that the distribution, when modeled as a von Mises-Fisher distribution, becomes more uniform as training progresses. There is experimental verification of this claim, and some results suggesting that the SNR is more correlated with their measure of uniformity than with the norm of the gradients.

Quality: The proofs appear correct to me.

Clarity: The paper is generally easy to read.

Originality & Significance: I don't know of this specific analysis existing in the literature, so in that sense it may be original. Nonetheless, I think there are serious issues with the significance. The idea that there are two phases of optimization is not particularly new (see for example Bertsekas 2015) and the paper's claim that uniformity of direction increases as SGD convergence is easy to see in a simple example. Consider f_i(x) = |x-b_i|^2  quadratics with different centers. Clearly the minimum will be the centroid. Outside of a ball of certain radius from the centroid all of the gradients grad f_i point in the same direction, closer to the minimum they will point towards their respective centers. It is pretty clear, then that uniformity goes up as convergence proceeds, depending on the arrangement of the centers.

The analysis in the paper is clearly more general and meaningful than the toy example, but I am not seeing what the take-home is other than the insight generated by the toy example. The paper would be improved by clarifying how this analysis provides additional insight, providing more analysis on the norm SNR vs uniformity experiment at the end.

Pros:
- SGD is a central algorithm and further analysis laying out its properties is important
- Thorough experiments.

Cons:
- It is not entirely clear what the contribution is.

Specific comments:
- The comment at the top of page 4 about the convergence of the minibatch gradients is a bit strange. This could also be seen as the reason that analysis of the convergence of SGD rely on annealed step sizes. Without annealing step-sizes, it's fairly clear that SGD will converge to a kind of stochastic process.

- The paper would be stronger if the authors try to turn this insight into something actionable, either by providing a theoretical result that gives guidance or some practical algorithmic suggestions that exploit it.

Dimitri P. Bertsekas. Incremental Gradient, Subgradient, and Proximal Methods for Convex Optimization: A Survey. ArXiv 2015.

---

> ### Author Response · Authors · 2018-11-12
> **Response to AnonReviewer2**
>
> We would like to clarify that we are not claiming to have discovered two-phase dynamics of SGD. As you have correctly pointed out, this behaviour has been known, and even more recently there have been a number of work analyzing this behaviour in deep learning. Li & Yuan (2017) investigated this behaviour by considering a shallow neural network with residual connections and assuming the standard normal input distribution and showed that SGD-based learning has two phases; (1) search and (2) convergence phases. Shwartz-Ziv & Tishby (2017) on the other hand investigated a deep neural network with tanh activation functions and showed that SGD-based learning has (1) drift (empirical error minimization, ERM) and (2) diffusion (representation compression) phases. Chee & Toulis (2018) instead looked at the inner product between successive minibatch gradients and presented transient and stationary phases. Our work investigates the dynamics of SGD-based learning in a perspective different from all these recent works by focusing more on geometry and directional analysis of minibatch gradients. In other words, our work provides yet another perspective on understanding the dynamics of SGD-based learning which is at the core of the recent success of deep learning.
>
> Furthermore, we would like to emphasize that our experiments, unlike most of the previous work, are conducted not only with neural networks that confirm well with theoretical assumptions but also with widely used neural networks (including latest techniques such as deep convolutional networks (Krizhevsky et al., 2015), residual networks (He et al., 2016) and batch normalization (Ioffe & Szegedy, 2015). These experiments revealed that our theoretical analysis applies well when all the latest techniques are used or/and when neural networks are shallow, providing some insight into SGD-based learning of generic neural networks.
>
> “The paper would be stronger if the authors try to turn this insight into something actionable”
>
> We strongly agree with you that it is desirable to find a practical algorithm based on the theoretical analysis. We however believe this is out of this paper’s scope, and we leave it as future research.
>
> - Li & Yuan (2017) Convergence analysis of two-layer neural networks with ReLU activation
> - Shwartz-Ziv & Tishby (2017) Opening the black box of deep neural networks via information
> - Chee & Toulis (2018) Convergence diagnostics for stochastic gradient descent with constant learning rate.
> - Krizhevsky et al., (2015) ImageNet classification with deep convolutional neural networks.
> - He et al., (2016) Deep residual learning for image recognition
> - Ioffe & Szegedy (2016) Batch normalization: Accelerating deep network training by reducing internal covariate shift.

---

### Official Review · AnonReviewer1 · 2018-11-03
**Interesting idea but implications, significance, theoretical analysis and experiments need improvements.**

**Rating:** 5
**Confidence:** 3

**Review:**


Quality and clarity: good.

Originality and significance: This paper studies the stochasticity
of the norms and directions of the mini-batch gradients, to
understand SGD dynamics. The contributions of this paper can be
summarized as: a) This paper defines gradient norm stochasticity as
the ratio of the variance of the stochastic norm to the expectation
of the stochastic norm. It theoretically and empirically shows that
this value is reduced as the batch size increases b) This paper
empirically finds that the distribution of angles between
mini-batch gradient and a given uniformly sampled unit vector
converges to an asymptotic distribution with mean 90 degrees, which
implies a uniform distribution of the mini-batch gradients. c)
This paper uses von Mises-Fisher Distribution to approximate the
distribution of the mini-batch gradients. By theoretically and
empirically observing that the estimated parameter \hat \kappa
decreases during training, they claim that the directional
uniformity of mini-batch gradients increases over SGD training.

The idea of measuring the uniformity of mini-batch gradients
through VMF distribution seems interesting. But it is unclear how
the study of this stochasticity dynamics of SGD can be related to
the convergence behavior of SGD for non-convex problems and/or the
generalization performance of SGD.

There are additional concerns/questions regarding both theoretical
part and empirical part:

[1] Section3.3: Assumption that p_i(w_0^0) =p_i(w_1^0) = p_i is not
reasonable when theoretically comparing \hat \kappa(w_1^0) and \hat
\kappa(w_0^0). The concentration parameter \hat \kappa(w) should be
estimated by the sum of the normalized mini-batch gradients "\hat
g_i(w)/||\hat g_i(w)||" . Instead of using mini-batch gradient,
this paper uses the sum of "p_i-w" by assuming that "p_i(w_0^0) -w"
is parallel to "\hat g_i(w)", which is ok. However, when comparing
\hat \kappa(w_0^0) and \hat \kappa(w_1^0), we say \hat
\kappa(w_0^0) = h(\sum p_i(w_0^0) - w_0^0) ) and \hat \kappa(w_1^0)
= h(\sum p_i(w_1^0) - w_1^0) ). It is not reasonable to use the
same p_i for p_i(w_0^0) and p_i(w_1^0) because p_i(w_0^0) -w_1^0 is
definitely not parallel to \hat g_i(w_1^0).

[2] Section 3.3: Assumption \hat g_i(w_t^{i-1}) \hat g_i(w_t^0) is
not convincing. With this assumption, the paper writes w_1^0 =
w_0^0 - \eta\sum_i \hat g_i(w_0^{i-1}) = w_0^0 - \eta\sum_i \hat
g_i(w_0^0) = w_0^0 - \eta \sum_i p_i-w_0^0. These equalities are
not persuasive. Because, \sum_i \hat g_i(w_0^0) is the full
gradient g(w_0^0) at w_0^0. In other words, these equalities imply
that from w_0^0 to w_1^0 (one epoch), SGD is doing a full gradient
descent: w_1^0 = w_0^0 -\eta g(w_0^0), which is not the case in
reality.

[3] Experiment: Batch size should be consistent with the given
assumption in the theoretical part. In theoretical part, \hat
\kappa(w_1^0) < \hat \kappa(w_0^0) is based on the assumption that
|\hat g_i(wt^{i-1}| \tat for all i, with *large mini-batch size*.
But in the experiment, they prove \hat \kappa(w_1^0) < \hat
\kappa(w_0^0) by using small-batch size which is 64. The authors
should either provide experiments with large batch size or try to
avoid the assumption of large batch size in theoretical part.

[4] The CNN experiment; It is better to add a discussion why the
\kappa increases in the early phase of training.

[5] The experiment results show, by the end of training, all models
FNN, DENN and CNN have very large value of \kappa which is around
10^4. This value implies that the mini-batch gradients distribution
is pretty concentrated, and it is contradictory to the statement in
the introduction which is "SGD converges or terminates when either
the norm of the minibatch gradient vanishes to zeros, or when the
angles of the mini-batch gradients are uniformly distributed and
their non-zero norms are close to each other''. It is also
contradictory to the experiment in 3.2 which implies the mini-batch
gradient are uniformly distributed after training.

[6] The notations in this paper can be improved, some notations are
using "i" for batch index, some notations are using "i" for one
data sample. Some notations in Section 3.3 and 3.1 can be moved to
Section 2 Preliminaries. It will be clearer to define all the
notations in one place.

Typos: -Section 3.1: first paragraph, E\hat g(w) -> E[\hat g(w)]; -
Paragraph before Lemma2: \hat \kappa increases -> \hat \kappa
decreases; - Paragraph after Theorem2: double the directions in "If
SGD iterations indeed drive the directions the directions of
minibatch gradients to be uniform".

---

> ### Author Response · Authors · 2018-11-12
> **Response to AnonReviewer1**
>
>
> “[1] Section3.3  It is not reasonable to use the same p_i for p_i(w_0^0) and p_i(w_1^0) because p_i(w_0^0) -w_1^0 is definitely not parallel to \hat g_i(w_1^0).”
>
> We believe the confusion may have arisen from our example in page 6 (together with Fig. 4b). Specifically, we want to clarify that we did not intend to say that \frac{\hat g_i(w_0^{i-1}}{\| \hat g_i(w_0^{i-1} \|} can be replaced with \frac{p_i-w_0^0}{\| p_i-w_0^0 \|}. Instead, our intention was to show that \sum_{i=1}^3 \frac{\hat g_i(w_0^{i-1}}{\| \hat g_i(w_0^{i-1} \|} could be replaced by \sum_{i=1}^3 \frac{p_i-w_0^0}{\| p_i-w_0^0 \|}, when all the sufficient conditions Corollary 3.1.
>
> “[2] Section 3.3: Assumption \hat g_i(w_t^{i-1}) [\approx] \hat g_i(w_t^0) is not convincing. “
>
> This is not an assumption we need. In fact, what we need is for the norms to be similar, i.e., \| \hat{g}_i(w_t^{i-1})\| \approx \| \hat{g}_i(w_t^0)\| for proving Corollary 3.1 (see Appendix D for its proof.)
>
> We state \hat g_i(w_t^{i-1}) \approx \hat g_i(w_t^0) as one simple possible case of having similar norms of these two minibatch gradients. We will clarify this in the next revision.
>
> “[3] Experiment: Batch size should be consistent with the given assumption in the theoretical part”
>
> The main theoretical analysis in our paper largely depends on the assumption that the norms of minibatch gradients are similar and not that the size of minibatch is large. We discuss a large size of minibatch as one case in which those norms are similar to each other, although there may be other ways for it to happen.
>
> We choose a reasonable minibatch size to confirm whether and in which setup our theoretical observation can be confirmed in practice. We however thank you for your suggestion and are running experiments while varying minibatch sizes at the moment. We will update the submission with new results soon.
>
> “[4] The CNN experiment; It is better to add a discussion why the \kappa increases in the early phase of training.”
>
> Our theoretical analysis makes a few assumptions on the loss function which is induced by the choice of a network architecture, such as the well-behavedness of the loss function. We conjecture our observation that the uniformity of minibatch gradients monotonically growing with a deep convolutional network equipped with residual connections and/or batch normalization is due to the fact that the loss function induced from this kind of network confirms well with the assumptions, as were discussed for instance earlier by Li & Yuan (2017) and Santurkar (2018). This is in contrast to deep networks without these latest techniques.
>
> - Li & Yuan (2017) Convergence analysis of two-layer neural networks with ReLU activation
> - Santurkar et al. (2018) How does batch normalization help optimization? (No, it is not about internal covariate shift)
>
> “[5] The experiment results show, by the end of training, all models FNN, DENN and CNN have very large value of \kappa which is around 10^4.”
>
> We would like to point out that the absolute value of $\kappa$ is not a good indicator of the uniformity due to its dependence on the dimensionality, as was investigated earlier by Cutting et al. Instead, our theoretical analysis and experiments focus on the trend of \kappa over training.
>
> - Cutting et al. [2017] Tests of concentration for low-dimensional and high-dimensional directional data
>
> “[6] The notations in this paper can be improved”
>
> Thanks for the suggestion! We will revise the text to make it clearer.

---

> > ### Author Response · Authors · 2018-11-15
> > **Additional response to AnonReviewer1 re estimated kappa values**
> >
> > “The experiment results show, by the end of training, all models FNN, DENN and CNN have very large value of \kappa which is around 10^4. This value implies that the mini-batch gradients distribution is pretty concentrated, and it is contradictory to the statement”
> >
> > In order to verify our earlier claim that the accuracy of the kappa estimate grows with respect to the dimensionality (the number of parameters of a neural network in our case), we have run some simulations. In these simulations, we vary the dimensionality, the number of samples and the true, underlying kappa by sampling from the vMF distribution with the designated kappa and estimating the kappa from these samples. We uploaded the plot from the simulation with 10,000 dimensions, which you can check from https://ibb.co/dNycc0 (both x- and y-axes are log10.) Unfortunately we could not easily go over 10,000 dimensions due to the difficulty in sampling from the vMF distribution.
> >
> > As can be seen from the uploaded plot, the estimated kappa approaches the true kappa from above as the number of samples increases. When the true kappa is large, the estimation error rapidly becomes zero as the number of samples approaches 3,000. When the true kappa is low (i.e., uniform over the angles), however, the gap does not narrow completely even with 3,000 samples.
> >
> > While fixing the true kappa to 0 and the number of samples to 3,000, we vary the dimensionality to empirically investigate the estimated kappa values. We chose to use 3,000 samples to be consistent with our experiments in the paper. We ran five simulations each and report both mean and standard deviation. See below for the estimates:
> >
> > d=200k, kappa=3,651.06+-5.10
> > d=640k, kappa=11,682.88+-3.57
> > d=2M,    kappa=36,526.49+-13.05
> >
> > We clearly observe the trend of increasing estimated kappas w.r.t. the dimensions. This suggests that we should not compare the absolute estimated kappas across different network architectures due to the differences in the number of parameters. This agrees well with Cutting et al. [2017] which empirically showed that the threshold for rejecting the null hypothesis of the true kappa being 0 grows with respect to the dimensions.
> >
> > At the end of training, DFNN+BN has approximately 2M parameters and the minimum estimated kappa we observed was 6.83 x 10^4, FNN+BN has approximately 640k parameters and the minimum estimated kappas was 2.04 x 10^4, and CNN+BN+Res has approximately 200k parameters and the minimum estimated kappas were 1.56 x 10^4. Considering these in the context of the simulation result above, we cannot say that the underlying directional distribution of minibatch gradients in all these cases at the end of training is not close to uniform.
> >
> > We again emphasize that our theory focuses more on the relative decrease of kappa (i.e. the relative increase of directional uniformity) rather than the absolute value of the estimated kappa.
> >
> > - Cutting et al. [2017] Tests of concentration for low-dimensional and high-dimensional directional data

---

> > > ### Comment · AnonReviewer1 · 2018-12-03
> > > **additional work needed**
> > >
> > > The authors have done a commendable job in addressing some of the concerns noted earlier.
> > >
> > > Given that the process is in a high-dimensional over-parameterized regime, it is still unclear if the explanations provided are still sufficient to capture the SGD dynamics. There is a body work which focus on the Hessian structure of the loss and effectively suggest that the sgd iterates live in a low-d space, e.g., see work by Sagun, Bottou, Lecun, and related papers. A high-d uniform random unit vector will, with high probability, be orthogonal to vectors living in a subspace. The observations made in the paper may not be capturing such sub-space concentration correctly -- or at least the claims need to be reconciled with existing observations regarding the Hessian being low rank. I encourage the authors to look into this carefully -- this can lead to considerably more clarity. Also, it will be good to consider the over-parameterized setting where the dimensionality d > number of samples n, which is typical in deep learning settings and where the stochastic gradients cannot span the full space.

---

> > > > ### Author Response · Authors · 2018-12-06
> > > > **Response to AnonReviewer1(additional work needed)**
> > > >
> > > > “A high-d uniform random unit vector will, with high probability, be orthogonal to vectors living in a subspace. The observations made in the paper may not be capturing such sub-space concentration correctly or at least the claims need to be reconciled with existing observations regarding the Hessian being low rank.”
> > > >
> > > > The increase in the uniformity of minibatch gradients in the original space implies the increasing uniformity in a lower-dimensional subspace. We however agree with the reviewer that it is desirable to study further the impact of the low-rank Hessian in learning a deep neural network. We leave this to the future.
> > > >
> > > >
> > > > “it will be good to consider the over-parameterized setting where the dimensionality d > number of samples n, which is typical in deep learning settings and where the stochastic gradients cannot span the full space”
> > > >
> > > > We agree with you that this is a more realistic and interesting setting, and thank you for your suggestion. We however would like to note that we assumed f_{I_i}(w) to be strictly convex to have a unique local minibatch solution p_i(w). In the over-parameterized setting where there are many zero eigenvalues in the hessian, we need a way to replace p_i(w) - w with a vector that characterizes the direction from w to a set of minibatch solutions, which is not trivial. We leave this as a future work.

---

### Official Review · AnonReviewer4 · 2018-11-08
**The theory looks good but how can it be used?**

**Rating:** 6
**Confidence:** 3

**Review:**


Gradient stochasticity is used to analyse the learning dynamics of SGD. It consists of two aspects: norm stochasticity and directional stochasticity. Although the norm stochasticity is easy to compute, it vanishes when the batch size increases. Therefore, it can be hard to measure the learning dynamics of SGD. The paper is motivated by measuring the learning dynamics by the directional stochasticity. Directly measuring the directional stochasticity with the ange distribution is hard, so the paper uses vMF distribution to approximate the uniformity measurement. The paper theoretically studies the proposed directional uniformity measurement. In addition, the experiments empirically show the directional uniformity measurement is more coherent with the gradient stochasticity.

1. As I’m not a theory person, I’m not very familiar with the related work on this line. But the analysis on the directional uniformity is interesting and original. So is the vMF approximation.
2. The theoretical analysis looks comprehensive and intuitive. And the authors did a reasonably good job on the experiments.
3. This paper provides some insights that warn people to pay attention to the directions of SGD. But the paper didn’t provide an answer on how this study can inform people to improve SGD. It’s true that the directional uniformity increases over training and it is correlated to the gradient. But what could this bring us remains unstudied.
4. Can the authors provide any theoretical or empirical analysis on why the directional uniformity didn’t increase in deep models like CNN and why it increases when BN and Res are applied?

---

> ### Author Response · Authors · 2018-11-12
> **Response to AnonReviewer4**
>
> “The paper didn’t provide an answer on how this study can inform people to improve SGD”
>
> We strongly agree with you and R2 that it is desirable to find a practical algorithm based on the theoretical analysis. We however believe this is out of this paper’s scope, and we leave it as future research.
>
> “Can the authors provide any theoretical or empirical analysis on why the directional uniformity didn’t increase in deep models like CNN and why it increases when BN and Res are applied?”
>
> Our theoretical analysis makes a few assumptions on the loss function which is induced by the choice of a network architecture, such as the well-behavedness of the loss function. We conjecture our observation that the uniformity of minibatch gradients monotonically growing with a deep convolutional network equipped with residual connections and/or batch normalization is due to the fact that the loss function induced from this kind of network confirms well with the assumptions, as were discussed for instance earlier by Li & Yuan (2017) and Santurkar (2018). This is in contrast to deep networks without these latest techniques.
>
> - Li & Yuan (2017) Convergence analysis of two-layer neural networks with ReLU activation
> - Santurkar et al. (2018) How does batch normalization help optimization? (No, it is not about internal covariate shift)

---

### Author Response · Authors · 2018-11-26
**Submission revised.**

Dear all Reviewers,

thanks for your valuable reviews and comments. According to your suggestions, we have made more discussions and experiments and updated the paper. In particular, we added/changed the following contents:

1) We have clarified the contribution of our paper and added a more detailed review of related work in the Introduction.
2) We have run experiments to see the effect of batch size on \kappa in Figure 5 in Section 4.1.
3) We have discussed the effect of adding batch normalization layers and residual connections in Section 4.2.
4) We have discussed briefly the estimated values of kappa near the end of training in Section 4.2. Details can be found in Supplemental Material G.
5) Some unclear parts have been re-written and they are better to read now.

We hope our responses can help address your concerns and questions.

---

### Meta-Review · Area_Chair1 · 2018-12-11
**Good quality and clarity but limited novelty and significance**

**Confidence:** 5
**Recommendation:** Reject

**Metareview:**

The paper presents a careful analysis of SGD by characterizing the stochastic gradient via von Mises-Fisher distributions. While the paper has good quality and clarity, and the authors' detailed response has further clarified several raised issues, some important concerns remain: Reviewer 1 would like to see careful discussions on related observations by other work in the literature, such as low rank Hessians in the over-parameterized regime, Reviewer 2 is concerned about the significance of the presented analysis and observations, and Reviewers 2 and 4 both would like to see how the presented theoretical analysis could be used to design improved algorithms. In the AC's opinion, while solid theoretical analysis of SGD is definitely valuable, it is highly desirable to demonstrate its practical value (considering that it does not provide clearly new insights about the learning dynamics of SGD).